# Serum metabolomic analysis of the dose-response effect of dietary choline in overweight male cats fed at maintenance energy requirements

**Alexandra Rankovic**[1], **Hannah Godfrey**[2], **Caitlin E. Grant**[2], **Anna K. Shoveller**[3], **Marica Bakovic**[4], **Gordon Kirby**[1], **Adronie Verbrugghe**[2]*

1 Department of Biomedical Sciences, Ontario Veterinary College, University of Guelph, Guelph, Ontario, Canada, 2 Department of Clinical Studies, Ontario Veterinary College, University of Guelph, Guelph, Ontario, Canada, 3 Department of Animal Biosciences, University of Guelph, Guelph, Ontario, Canada, 4 Department of Human Health and Nutritional Sciences, University of Guelph, Guelph, Ontario, Canada

* averbrug@uoguelph.ca

**Data Availability Statement:** All data files are available from Borealis, the Canadian Dataverse

## Abstract

Choline participates in methyl group metabolism and has been recognized for its roles in lipid metabolism, hepatic health and muscle function in various species. Data regarding the impacts of choline on feline metabolic pathways are scarce. The present study investigated how choline intake affects the metabolomic profile of overweight cats fed at maintenance energy. Overweight (n = 14; body condition score:6-8/9) male adult cats were supplemented with five doses of choline in a 5x5 Latin Square design. Cats received a daily dose of choline on extruded food (3620 mg choline/kg diet) for three weeks at maintenance energy requirements (130 kcal/kgBW0.4). Doses were based on body weight (BW) and the daily recommended allowance (RA) for choline for adult cats (63 mg/kg $BW^{0.67}$). Treatment groups included: Control (no additional choline, 1.2 x NRC RA, 77 mg/kg $BW^{0.67}$), 2 x NRC RA (126 mg/kg $BW^{0.67}$), 4 x NRC RA (252 mg/kg $BW^{0.67}$), 6 x RA (378 mg/kg $BW^{0.67}$), and 8 x NRC RA (504 mg/kg $BW^{0.67}$). Serum was collected after an overnight fast at the end of each treatment period and analyzed for metabolomic parameters through nuclear magnetic resonance (NMR) spectroscopy and direct infusion mass spectrometry (DI-MS). Data were analyzed using GLIMMIX, with group and period as random effects, and dose as the fixed effect. Choline up to 8 x NRC RA was well-tolerated. Choline at 6 and 8 x NRC RA resulted in greater concentrations of amino acids and one-carbon metabolites (P < 0.05) betaine, dimethylglycine and methionine. Choline at 6 x NRC RA also resulted in greater phosphatidylcholine and sphingomyelin concentrations (P < 0.05). Supplemental dietary choline may be beneficial for maintaining hepatic health in overweight cats, as it may increase hepatic fat mobilization and methyl donor status. Choline may also improve lean muscle mass in cats. More research is needed to quantify how choline impacts body composition.

Repository, for the University of Guelph (https://doi.org/10.5683/SP3/QFVIFF).

**Funding:** This research was supported by a Natural Sciences and Engineering Research Council (NSERC) Collaborative Research and Development grant (CRDPJ #472710-16), in partnership with Elmira Pet Products (Elmira, ON, Canada). The choline chloride supplement was kindly provided by Balchem (New Hampton, NY, United States of America). The funders had no role in the study design, data collection and analysis, decision to publish, or preparation of the present manuscript.

**Competing interests:** The authors declare no conflicts of interest. AR and HG declare that they have participated in paid internships and engagements with pet food companies within Canada. CG holds the Nestle Purina Professorship in Companion Animal Nutrition at the Ontario Veterinary College, is the owner of Grant Veterinary Nutrition Services and consults with Simmons Pet Food. AKS is the Champion Petfoods Chair in Canine and Feline Nutrition, Physiology and Metabolism, consults for Champion Petfood, was previously employed by P&G and Mars Pet Care, serves on the Scientific Advisory Board for Trouw Nutrition, and has received honoraria and research funding from various commodity groups, pet food manufacturers, and ingredient suppliers. AV is the Royal Canin Veterinary Diets Endowed Chair in Canine and Feline Clinical Nutrition and declares that they serve on the Health and Nutrition Advisory Board for Vetdiet. AV has also received honoraria and research funding from various pet food manufacturers and ingredient suppliers. At the time of the study, the Ontario Veterinary College received funding from Nestlé Purina Proplan Veterinary Diets to support a Registered Veterinary Technician in Clinical Nutrition, who helped perform the described animal trial. This does not alter our adherence to PLOS ONE policies on sharing data and materials.

## Introduction

Obesity is a major concern for many species due to devastating associated secondary health conditions and cats are no exception [1–3]. Obesity puts cats at high risk of various diseases, including diabetes mellitus, osteoarthritis and lower urinary tract diseases [4–6]. Ultimately, obesity can reduce the overall quality of a cat's life [7]. It is estimated that up to 63% of cats are considered overweight or obese [4, 5, 8–16], depending on the country and the criteria used to classify and assess adiposity. Although weight loss programs are recommended for these cats, weight loss in cats can be a long and difficult journey [17, 18]. Low degrees of dietary energy restriction often do not result in weight loss in obese cats, and a higher degree of energy restriction may therefore be required [19, 20]. However, if too high a degree of energy restriction (estimated to be between 50–75%) is placed on an obese cat, they could be at risk of developing feline hepatic lipidosis (FHL) [21].

Feline hepatic lipidosis is characterized by an overaccumulation of triglycerides (TAG) within the liver [22]. Triglycerides accounted for 34% of the liver mass in cats with FHL, as compared to 1% in the healthy control cats [23]. Although it is considered one of the most common liver diseases affecting cats [24, 25], the pathogenesis of FHL is not fully understood. A period of drastic energy restriction or anorexia in obese cats is believed to be the primary event in the development of FHL [22]. Cats with FHL had hepatic fatty acid profiles that resembled the fatty acid composition of their adipose tissue [23]. As a result, current evidence leads to the hypothesis that fatty acids from adipose tissue are mobilized to the liver.

Of interest for the prevention and treatment of FHL is the nutrient choline. Although some endogenous synthesis does occur, choline is considered an essential nutrient for cats [26–28]. The National Research Council's (NRC) recommended allowance (RA) for dietary choline in adult cats is 63 mg/kg $BW^{0.67}$ [29]. Choline is involved in many metabolic pathways, including its role in one-carbon metabolism through its derivative betaine, and in lipid metabolism through phosphatidylcholine (PC) [30–32]. Betaine functions as one of the methyl group donors that can facilitate the re-methylation of homocysteine to produce methionine. Subsequently, the production of methionine can lead to the production of s-adenosylmethionine (SAMe) [33, 34]; a universal methyl donor required for DNA and protein methylation and synthesis of endogenous metabolites, including L-carnitine [35]. The entry of fatty acids into the mitochondria for β-oxidation relies on L-carnitine [36]. Phosphatidylcholine is a necessary lipid component of very-low-density lipoproteins (VLDL), required to mobilize TAG out of the liver and into circulation [37]. Choline deficiency leading to fatty liver has been documented in cats [26, 27]. Fatty liver is similarly a common outcome of choline deficiency in many other species, including humans and rodents [38–43]. This is due to the inability of the liver to synthesize and excrete VLDL, and subsequently TAG, without PC [44].

Given the metabolic pathways that choline participates in, it has been proposed that choline supplementation may prove useful in the prevention of FHL [45]. However, before choline supplementation can be assessed as a possible nutritional intervention for obese cats undergoing weight loss, an adequate dose must be determined. Previous research by Verbrugghe et al. [46] and Rankovic et al. [47] found that choline at 5 x NRC RA in obese cats and choline at 6 x NRC RA in overweight cats, respectively, appeared to improve hepatic lipid mobilization as increased serum TAG, cholesterol (CHOL), high-density lipoprotein cholesterol (HDL-C) and VLDL were observed in both studies. In-depth analyses of choline-related metabolic pathways were however not performed in these studies. Metabolomics research in growing kittens consuming choline at 3 x NRC RA for 12 weeks revealed increased serum one-carbon metabolites and decreased medium-chain acylcarnitines, as compared to control kittens consuming choline at 0.8 x RA [48]. Choline supplementation above the NRC recommendations could benefit

one-carbon metabolism and fatty acid oxidation. As said study was performed during the growth phase [48], it remains unclear how choline supplementation may affect one-carbon and lipid metabolism in adult cats, and those with an increased liver TAG, as seen in obese cats [49]. Additionally, the dose at which choline would most benefit both PC synthesis and one-carbon metabolism in cats has yet to be determined.

The application of serum metabolomics allows for a comprehensive view of complex metabolic pathways, and how these pathways are altered by dietary choline intake. By quantifying low molecular weight metabolites in serum through quantitative nuclear magnetic resonance (NMR) spectroscopy and direct flow injection mass spectrometry (DI-MS), the present study aimed to establish changes in biochemical pathways, resulting from different doses of dietary choline supplementation in overweight male adult cats.

## Materials and methods

The University of Guelph Animal Care Committee (AUP#4118) approved all procedures following provincial and national animal care and use guidelines [47].

### Animals

Domestic shorthair (DSH) male neutered cats (n = 14) were enrolled. Cats underwent physical examinations, complete blood counts (CBC) and serum biochemistry profiles, to determine health status. Cats were 1 year of age at the start of the trial and had a mean body weight (BW) of 4.97 ± 0.16 kg (range: 4.36–6.24 kg). The cats had body condition scores (BCS) of 6 or greater out of 9 (mean ± SEM = 6.87 ± 0.18; range = 6–8) [50].

### Housing

Cats were housed together in an indoor free-living environment (23 ft x 19 ft) at the Animal Biosciences Cattery at the Ontario Agricultural College of the University of Guelph (Guelph, ON, Canada). Various sources of enrichment were provided within the room, including toys, perches, scratching posts, cat trees, and boxes for hiding. Cats also received up to two hours of human interaction with familiar people five days a week. This interaction included voluntary play with high-value toys, petting and brushing. Cats had *ad libitum* access to distilled water within the room.

Surfaces in the room were cleaned and sanitized daily. Litterboxes were cleaned twice daily and topped up with litter as needed. Lights were turned on at 0700 h and turned off at 1900 h, for a controlled 12 h light 12 h dark cycle within the room. Temperature and humidity were maintained at 23˚C and 40%, respectively.

### Diet and feeding

Four weeks before the start, and throughout the trial, cats were fed a commercial extruded diet (Nutram Total Grain-Free® Chicken and Turkey Recipe, Elmira Pet Products, Elmira, ON, Canada), formulated for adult maintenance according to the Association of American Feed Control Officials (AAFCO). Nutrient analyses of the diet, including proximate analysis, vitamins, minerals and amino acids, were previously described by Rankovic et al. [47]. The diet contained 3901 mg choline/kg diet dry matter basis (DMB), as determined by the enzymatic colorimetric method, described by the Association of Official Analytical Chemists (AOAC 999.14) [51].

Cats were fed to maintain their current BWs. The initial quantity of food was calculated using the following equation for overweight cats at maintenance: 130 kcal/kg $BW^{0.4}$ [29]. Body

condition score and BW were assessed and recorded weekly. Individual food quantities were adjusted as needed throughout the trial.

## Choline supplementation and study design

The cats were separated into five groups prior to the trial. Choline chloride (Pet Shure, 97% Choline Chloride, 72.3% choline; Balchem Corporation, New Hampton, NY, United States of America) was top-dressed onto the food in a 5 x 5 Latin Square design. The choline doses were based on individual metabolic BW ($BW^{0.67}$) and the NRC recommended allowance (RA) for choline (63 mg/kg $BW^{0.67}$) for adult cats, and took into account the estimated choline intake from the base diet. The five doses were: control (no additional choline added, 1.2 x NRC RA, 77 mg/kg $BW^{0.67}$), 2 x NRC RA (126 mg/kg $BW^{0.67}$), 4 x NRC RA (252 mg/kg $BW^{0.67}$), 6 x NRC RA (378 mg/kg $BW^{0.67}$), and 8 x NRC RA (504 mg/kg $BW^{0.67}$). Choline was provided daily for a period of three weeks per dose. The choline chloride was first dissolved in distilled water to form a stock solution (549.62 ± 16.23 mg choline/mL distilled water). The choline solution was pipetted and stored in cryovials (Fisherbrand Premium Microcentrifuge Tubes: 1.5mL) in one-week increments, based on the cats' individual BWs.

Cats were separated once daily at 0800 h, to be fed individually. The daily food for each cat was divided into two allotments: ¼ of their daily food intake, and the remaining ¾. The first allotment (¼ daily food intake) was top-dressed with the pre-measured choline chloride solution and left to soak for 20 minutes before feeding. Once the first allotment with the choline was consumed, each cat received the remainder of its food (¾ daily food intake). Cats had up to one hour each to consume their food. Individual orts were measured and recorded daily.

## Blood collection and laboratory analyses

All cats were fasted before blood collection (23 hours since last meal). Dexmedetomidine hydrochloride (Dexdomitor, Zoetis, Kirkland, QC, Canada) (0.5 mg/ml) was given intramuscularly at a dose of 0.01 mg/kg BW. Whole blood was collected 20 minutes following sedation [52]. Atipamezole (Antisedan, Zoetis, Kirkland, QC, Canada) (5 mg/ml) was administered intramuscularly at a dose of 0.1 mg/kg BW to reverse sedation [53].

Whole blood (5 mL) was sampled via venipuncture (BD Precision Glide™ Needles 23G x, Becton Dickson, Franklin Lakes, NJ, United States of America) from the jugular vein of 14 cats. Whole blood was consistently collected from the medial saphenous vein of one cat. Blood was stored in serum separating tubes (BD Vacutainer™ Venous Blood Collection Tubes: Serum Separating Tubes: Hemogard, Becton Dickson, Franklin Lakes, NJ, United States of America) at 5°C until centrifugation. Whole blood was centrifuged at 2500 g x 15 min at 4°C (LegendRT, Kendro Laboratory Products 2002, Germany) within 2 hours of sampling. Serum was separated and aliquoted into 1.5 mL cryovials. Cryovials were stored at -20°C until analysis. Samples were shipped on dry ice to The Metabolomics Innovation Center (TMIC) at the University of Alberta (Edmonton, AB, Canada), where quantitative NMR spectroscopy and DI-MS were performed.

Before NMR spectroscopy, an initial deproteinization step was applied to the serum samples, following the procedures described by Psychogios et al. [54]. Serum was then centrifuged (10, 000 rpm x 20 minutes) and spectral analysis of samples (250 μL) was performed using an Bruker Avance III 700 MHz NMR spectrometer equipped with a 5 mm cryoprobe (Bruker Corporation, Billerica, MA, United States of America) [55]. Automated spectral analysis of raw NMR data was performed using an in-house version of the magnetic resonance for metabolomics (MAGMET: http://magmet.ca/users/login) software package and a custom metabolite library. A total of 52 water-soluble plasma metabolites were identified and classified by

metabolic pathway using the Human Metabolome Database (HMDB: http://www.hmdb.ca) into: gluconeogenic amino acids, ketogenic amino acids, gluconeogenic and ketogenic amino acids, amino acid degradation products, glycolysis, TCA cycle, ketogenesis, one-carbon metabolism, purine degradation products, alcohols, and other.

Additionally, endogenous metabolites were determined by the combination of DI-MS with reverse-phase liquid chromatography (LC)-tandem mass spectrometry (MS/MS) custom assay, as previously described by Ren et al. [56]. Mass spectrometric analysis of serum samples (100 μL) was done using a 4000 Qtrap mass spectrometer (Applied Biosystems/MDS Analytical Technologies, Foster City, CA, United States of America) with an Agilent 1260 series ultra-high performance liquid chromatography (UHPLC) system (Agilent Technologies, Palo Alto, CA. United States of America). The obtained 135 metabolites were similarly grouped by class using HMDB into: biogenic amines; amino acids, amino acid derivatives and ammonium compounds; acylcarnitines; phosphatidylcholine diacyl (PC aa) and phosphatidylcholine acyl-alkyl (PC ae); lysophosphatidylcholines (LPC); sphingomyelins (SM) and hydroxysphingo-myelins (HSM); and organic acids and sugars. Totals for acylcarnitines, PC aa, PC ae, LPC, HSM, and SM were calculated as a sum of the respective metabolites within that group. In addition to total acylcarnitines, total short-chain acylcarnitines (sum of C2 through C5), total medium-chain acylcarnitines (sum of C6 through C12) and total long-chain acylcarnitines (sum of C14 through C18) were also calculated.

## Statistical analyses

Data was analyzed using Statistical Analysis System (SAS® Studio, 3.8, SAS Institute, Cary, NC, United States of America). Studentized residuals were assessed for normality by scatter plots via visual assessment and the Shapiro-Wilk test, for both NMR and DI-MS. A lognormal distribution was applied when residuals were not normally distributed (NMR: L-alanine, 2-hydroxybutyric acid, succinate, ornithine, acetone, acetoacetate, creatine, methanol, ethanol, isopropanol, and 2-hydroxyisovalerate; DI-MS: histamine, alanine, C3, C4, C4:1, C6, C7:DC, C8, C12, C14, C18, total medium-chain acylcarnitines, fumaric acid, and isobutyric acid). Where a lognormal distribution was specified, the data was back-transformed to obtain the least square mean (LSM) for each response variable.

Analyses of all NMR and DI-MS data were done using the GLIMMIX covariant analysis procedure. Cat was used as the subject, dose as the fixed effect, and period and group as the random effects. The covariance structure with the smallest Akaike information criterion (AIC) value was applied. Results are expressed as least square mean (LSM) ± standard error of the mean (SEM). Significance was considered at P < 0.05. A P-value of < 0.10 was considered a trend. A Tukey's post hoc test was applied to separate and compare means where a significant effect of dose was present.

Metaboanalyst 5.0 was used to create heatmaps for the NMR and DI-MS data. Mean centering was selected for data scaling for both DI-MS and NMR data. Heatmaps were created with Euclidean distance measures and clustered with the Ward algorithm.

## Results

### NMR

Metabolites determined through quantitative NMR spectroscopy are presented in Table 1. Additionally, the heatmap of the nine NMR metabolites with a significant effect of dose are presented in Fig 1.

Choline significantly affected two of the metabolites involved in one-carbon metabolism: betaine and dimethylglycine (DMG) ($P_{Dose} < 0.0001$, and $P_{Dose} \leq 0.0001$, respectively). Serum

**Table 1. Mean serum concentrations (μM) of metabolites determined by quantitative NMR spectroscopy in overweight cats (n = 14) receiving control (no additional choline supplementation, 1.2 x NRC RA, 77 mg/kg BW$^{0.67}$), choline at 2 x NRC RA (126 mg/kg BW$^{0.67}$), 4 x NRC RA (252 mg/kg BW$^{0.67}$), 6 x NRC RA (378 mg/kg BW$^{0.67}$), and 8 x NRC RA (504 mg/kg BW$^{0.67}$), in a 5 x 5 Latin square design for 3-week periods.**

| | | Choline Dose | | | | | |
| --- | --- | --- | --- | --- | --- | --- | --- |
| | Metabolite | Control | 2 x NRC RA | 4 x NRC RA | 6 X NRC RA | 8 X NRC RA | P$_{Dose}$ |
| Gluconeogenic Amino Acids | Glycine | 343.98 ± 15.38 | 359.13 ± 14.96 | 360.27 ± 14.60 | 374.86 ± 14.60 | 365.14 ± 14.60 | 0.454 |
| | L-Alanine | 443.90 ± 50.09 | 471.95 ± 51.80 | 481.44 ± 51.55 | 570.01 ± 61.03 | 519.45 ± 55.61 | 0.254 |
| | L-Arginine | 159.72 ± 8.47 | 170.71 ± 8.28 | 171.34 ± 8.11 | 176.90 ± 8.11 | 181.37 ± 8.11 | 0.113 |
| | L-Asparagine | 51.59 ± 4.07[ab] | 46.05 ± 4.03[b] | 54.75 ± 3.99[a] | 56.71 ± 3.99[a] | 57.39 ± 3.99[a] | **0.0007** |
| | L-Aspartic Acid | 61.21 ± 7.27 | 63.55 ± 7.11 | 67.06 ± 6.97 | 72.24 ± 6.97 | 71.94 ± 6.97 | 0.386 |
| | L-Glutamic acid | 102.02 ± 15.02 | 119.92 ± 14.90 | 107.43 ± 14.79 | 112.52 ± 14.79 | 101.64 ± 14.79 | 0.314 |
| | L-Glutamine | 748.08 ± 35.83 | 701.64 ± 35.03 | 758.13 ± 34.32 | 699.12 ± 34.32 | 723.01 ± 34.32 | 0.282 |
| | L-Methionine | 44.83 ± 2.47[b] | 45.03 ± 2.43[b] | 49.11 ± 2.39[ab] | 50.43 ± 2.39[a] | 51.26 ± 2.39[a] | **0.005** |
| | L-Proline | 126.90 ± 6.92[b] | 141.47 ± 6.71[ab] | 149.47 ± 6.53[a] | 157.23 ± 6.53[a] | 149.46 ± 6.53[a] | **0.004** |
| | L-Serine | 140.20 ± 6.18[b] | 139.55 ± 6.02[b] | 157.15 ± 5.88[ab] | 170.55 ± 5.88[a] | 160.52 ± 5.88[a] | **< 0.0001** |
| | L-Valine | 159.30 ± 7.15 | 148.96 ± 7.03 | 158.01 ± 6.93 | 152.32 ± 6.93 | 151.06 ± 6.93 | 0.316 |
| | Ornithine | 33.51 ± 5.37 | 29.42 ± 4.57 | 45.91 ± 6.93 | 39.15 ± 5.91 | 35.90 ± 5.42 | 0.128 |
| Ketogenic Amino Acids | L-Leucine | 111.85 ± 6.35 | 108.91 ± 6.24 | 111.82 ± 6.13 | 111.91 ± 6.13 | 108.39 ± 6.13 | 0.925 |
| | L-Lysine | 157.32 ± 8.46[b] | 165.06 ± 8.23[ab] | 172.41 ± 8.03[ab] | 169.54 ± 8.03[ab] | 188.76 ± 8.03[a] | **0.014** |
| Gluconeogenic & Ketogenic Amino Acids | L-Isoleucine | 51.33 ± 3.18 | 55.35 ± 3.10 | 54.34 ± 3.02 | 54.38 ± 3.02 | 51.71 ± 3.02 | 0.680 |
| | L-Phenylalanine | 77.44 ± 4.41 | 81.00 ± 4.34 | 79.88 ± 4.28 | 85.01 ± 4.28 | 81.78 ± 4.28 | 0.336 |
| | L-Threonine | 117.00 ± 4.75[b] | 118.43 ± 4.76[ab] | 118.46 ± 4.59[ab] | 128.73 ± 4.43[ab] | 134.56 ± 4.43[a] | **0.012** |
| | L-Tyrosine | 49.07 ± 3.42 | 48.42 ± 3.37 | 49.75 ± 3.32 | 50.60 ± 3.32 | 51.17 ± 3.32 | 0.835 |
| Amino Acid Degradation Products | 2-Hydroxybutyric Acid | 12.18 ± 1.41 | 11.46 ± 1.31 | 12.11 ± 1.36 | 12.26 ± 1.37 | 11.26 ± 1.26 | 0.855 |
| | Creatinine | 133.61 ± 5.55 | 138.33 ± 5.37 | 140.32 ± 5.21 | 152.07 ± 5.21 | 145.47 ± 5.21 | 0.063 |
| Glycolysis | Acetic Acid | 28.33 ± 6.14 | 31.27 ± 6.03 | 34.46 ± 5.94 | 39.52 ± 5.94 | 29.56 ± 5.94 | 0.217 |
| | D-Glucose | 11042.38 ± 1071.44 | 10249.68 ± 1047.23 | 10019.55 ± 1025.66 | 10141.64 ± 1025.63 | 10550.71 ± 1025.66 | 0.867 |
| | L-Lactic Acid | 1305.10 ± 149.47 | 1453.92 ± 147.49 | 1197.32 ± 145.73 | 1315.66 ± 145.73 | 1415.50 ± 145.73 | 0.131 |
| TCA Cycle | Citric Acid | 210.64 ± 12.81 | 197.27 ± 12.43 | 209.22 ± 12.11 | 213.04 ± 12.11 | 207.13 ± 12.11 | 0.829 |
| | Malonate | 11.51 ± 1.26 | 10.28 ± 1.22 | 9.38 ± 1.18 | 12.99 ± 1.18 | 11.62 ± 1.18 | 0.171 |
| | Oxoglutarate | 6.35 ± 1.57 | 5.83 ± 1.54 | 4.42 ± 1.51 | 6.74 ± 1.51 | 6.46 ± 1.51 | 0.507 |
| | Pyruvic Acid | 17.04 ± 4.16 | 12.56 ± 4.02 | 12.92 ± 4.16 | 13.32 ± 4.16 | 12.56 ± 4.16 | 0.879 |
| | Succinate | 1.42 ± 0.13 | 1.62 ± 0.15 | 1.52 ± 0.13 | 1.50 ± 0.13 | 1.48 ± 0.12 | 0.750 |
| Ketogenesis | Acetone | 8.54 ± 1.48 | 7.32 ± 1.22 | 7.50 ± 1.20 | 8.27 ± 1.33 | 6.05 ± 0.97 | 0.579 |
| | Acetoacetate | 0.78 ± 0.27 | 0.83 ± 0.24 | 0.91 ± 0.26 | 0.52 ± 0.17 | 0.56 ± 0.16 | 0.561 |
| | 3-Hydroxybutyric Acid | 29.33 ± 2.57 | 27.59 ± 2.53 | 30.30 ± 2.48 | 24.67 ± 2.48 | 25.93 ± 2.48 | 0.066 |
| One Carbon Metabolism | Betaine | 256.13 ± 59.06[c] | 366.27 ± 57.05[c] | 606.40 ± 55.30[b] | 845.17 ± 55.30[a] | 834.56 ± 55.30[a] | **< 0.0001** |
| | Choline | 19.73 ± 3.56 | 19.54 ± 3.49 | 22.71 ± 3.43 | 22.32 ± 3.43 | 21.17 ± 3.43 | 0.874 |
| | Creatine | 9.67 ± 0.88 | 9.71 ± 0.85 | 10.53 ± 0.89 | 10.78 ± 0.91 | 11.78 ± 0.99 | 0.432 |
| | Dimethylglycine | 9.07 ± 0.98[b] | 9.88 ± 0.96[b] | 11.29 ± 0.93[b] | 15.10 ± 0.93[a] | 14.94 ± 0.93[a] | **< 0.0001** |
| | Formic Acid | 25.36 ± 1.87 | 26.31 ± 1.81 | 27.00 ± 1.76 | 29.33 ± 1.76 | 25.42 ± 1.76 | 0.353 |
| | L-carnitine | 26.94 ± 3.75 | 29.37 ± 3.60 | 32.59 ± 3.47 | 37.14 ± 3.47 | 25.34 ± 3.47 | 0.116 |
| Purine Degradation | Hypoxanthine | 17.52 ± 3.23 | 17.91 ± 3.12 | 15.62 ± 3.01 | 25.89 ± 3.01 | 20.52 ± 3.01 | 0.153 |
| Alcohols | Methanol | 1139.02 ± 589.97 | 518.77 ± 254.58 | 674.69 ± 318.35 | 1889.82 ± 891.65 | 504.90 ± 238.23 | 0.153 |
| | Ethanol | 94.62 ± 32.11 | 52.28 ± 16.96 | 61.90 ± 19.33 | 108.80 ± 33.94 | 86.45 ± 26.99 | 0.447 |
| | Isopropanol | 426.11 ± 271.27[b] | 115.35 ± 72.11[ab] | 141.27 ± 86.89[ab] | 87.87 ± 53.98[a] | 223.21 ± 137.27[ab] | **0.044** |
| | Propylene Glycol | 161.77 ± 22.75 | 203.06 ± 21.75 | 172.99 ± 20.96 | 195.67 ± 20.90 | 168.91 ± 20.96 | 0.681 |

(*Continued*)

**Table 1.** (Continued)

| | Metabolite | Choline Dose | | | | | P<sub>Dose</sub> |
|---|---|---|---|---|---|---|---|
| | | Control | 2 x NRC RA | 4 x NRC RA | 6 X NRC RA | 8 X NRC RA | |
| Other | D-Mannose | 36.44 ± 3.93 | 37.31 ± 3.78 | 38.44 ± 3.64 | 39.20 ± 3.64 | 38.78 ± 3.64 | 0.984 |
| | L-Acetyl Carnitine | 2.21 ± 0.32 | 2.33 ± 0.31 | 1.93 ± 0.30 | 1.96 ± 0.30 | 2.16 ± 0.30 | 0.869 |
| | 2-Hydroxyisovalerate | 4.07 ± 0.76 | 3.20 ± 0.58 | 4.03 ± 0.70 | 4.34 ± 0.75 | 3.18 ± 0.55 | 0.611 |
| | L-Alpha aminobutyric acid | 0.81 ± 0.37 | 1.00 ± 0.35 | 0.76 ± 0.34 | 1.03 ± 0.35 | 1.12 ± 0.34 | 0.922 |
| | 3-Methyl 2-oxovaleric acid | 3.78 ± 0.48 | 3.25 ± 0.46 | 3.39 ± 0.44 | 3.87 ± 0.44 | 3.07 ± 0.44 | 0.639 |
| | Ketoleucine | 3.85 ± 0.54 | 3.80 ± 0.52 | 3.61 ± 0.51 | 4.39 ± 0.51 | 3.86 ± 0.51 | 0.599 |
| | 3-Hydroxyisovalerate | 2.33 ± 0.57 | 1.74 ± 0.55 | 1.68 ± 0.54 | 1.80 ± 0.54 | 2.81 ± 0.54 | 0.377 |
| | Dimethylamine | 8.85 ± 0.89 | 8.18 ± 0.88 | 7.40 ± 0.87 | 8.48 ± 0.87 | 7.80 ± 0.87 | 0.113 |
| | Dimethylsulfone | 4.33 ± 0.41 | 4.25 ± 0.40 | 4.84 ± 0.38 | 4.73 ± 0.38 | 4.70 ± 0.38 | 0.762 |
| | Urea | 3220.27 ± 235.93 | 2905.38 ± 227.55 | 2865.08 ± 220.22 | 3367.11 ± 220.21 | 2961.26 ± 220.22 | 0.317 |

Values expressed as LSM ± SEM; Values in a row with superscripts without a common letter differ; P < 0.05, Repeated measures ANOVA with Tukey post-hoc test.

NMR = nuclear magnetic resonance; BW = body weight; NRC = National Research Council; RA = Recommended Allowance.

concentrations of both betaine and DMG increased with choline at 6 x and 8 x NRC RA, as compared to control, 2 x and 4 x NRC RA. Additionally, choline at 4 x resulted in greater concentrations of betaine, as compared to control and 2 x NRC RA. Choline dose did not impact the other one-carbon metabolites investigated: choline, creatine, formic acid and L-carnitine ($P_{Dose}$ = 0.874, 0.432, 0.353, and 0.116, respectively).

The gluconeogenic amino acids L-asparagine, L-proline, L-methionine and L-serine were significantly affected by dose ($P_{Dose}$ = 0.0007, $P_{Dose}$ = 0.004, $P_{Dose}$ = 0.005, and $P_{Dose} \leq 0.0001$, respectively). L-asparagine increased with 4 x, 6 x, and 8 x NRC RA, as compared to 2 x NRC

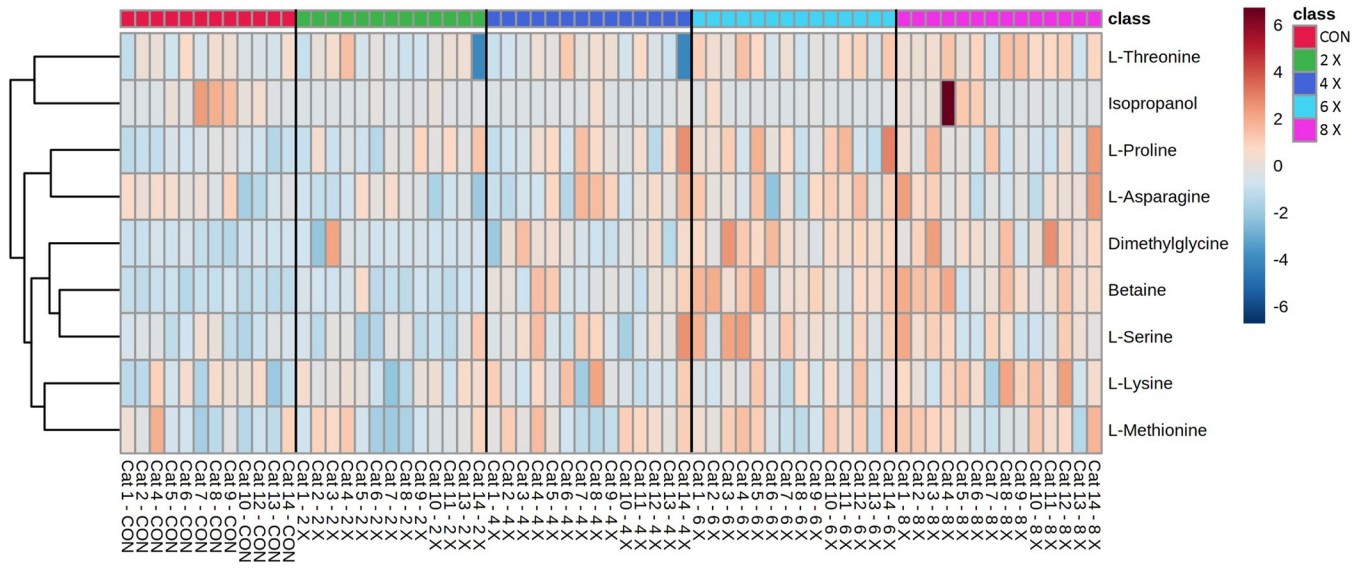

**Fig 1. Heatmap with euclidean distance and ward clustering of mean serum metabolites determined by quantitative NMR spectroscopy in overweight cats (n = 14) receiving control (no additional choline supplementation, 1.2 x NRC RA, 77 mg/kg BW0.67), choline at 2 x NRC RA (126 mg/kg BW0.67), 4 x NRC RA (252 mg/kg BW0.67), 6 x NRC RA (378 mg/kg BW0.67), and 8 x NRC RA (504 mg/kg BW0.67), in a 5 x 5 Latin square design for 3-week periods, with a significant effect of choline dose (P < 0.05).**

RA. Similarly, serum concentrations of L-proline were greater with 4 x, 6 x, and 8 x NRC RA, as compared to control. Both L-methionine and L-serine were increased with choline supplementation at 6 x, and 8 x NRC RA as compared to control and 2 x NRC RA. There was no effect of choline dose on the remaining gluconeogenic amino acids, including glycine, L-alanine, L-arginine, L-aspartic acid, L-glutamic acid, L-glutamine, L-valine and ornithine ($P_{Dose} > 0.10$).

The serum concentrations of the ketogenic amino acid L-lysine increased with choline at 8 x NRC RA, as compared to control ($P_{Dose} = 0.014$). L-leucine was not affected by choline dose ($P_{Dose} = 0.925$). Choline supplementation at 8 x NRC RA increased the gluconeogenic and ketogenic amino acid L-threonine, compared to control ($P_{Dose} = 0.012$). Choline did not affect serum concentrations of the remaining gluconeogenic and ketogenic amino acids: L-isoleucine, L-phenylalanine, and L-tyrosine ($P_{Dose} = 0.680, 0.336,$ and $0.835$, respectively). There was no change in the serum concentration of the amino acid degradation product 2-hydroxybutyric acid ($P_{Dose} = 0.855$). A trend was observed for serum creatinine ($P_{Dose} = 0.063$).

No significant differences were noted in the serum concentrations of metabolites involved in glycolysis, the TCA cycle, or ketogenesis, as analyzed by NMR spectroscopy ($P_{Dose} > 0.05$). A trend was observed for serum 3-hydroxybutyric acid ($P_{Dose} = 0.066$).

Serum isopropanol was lower with choline at 6 x NRC RA as compared to control ($P_{Dose} = 0.044$). The serum concentrations of the other alcohols analyzed by NMR spectroscopy; methanol, ethanol and isopropanol, did not change with choline dose ($P_{Dose} = 0.153, 0.447,$ and $0.681$, respectively). Similarly, choline dose did not alter serum hypoxanthine ($P_{Dose} = 0.153$), or any of the other metabolites determined by NMR spectroscopy ($P_{Dose} > 0.10$).

## DI-MS

Metabolites determined through DI-MS are presented in Tables 2–8. Heatmaps of the 47 DI-MS metabolites with a significant effect of dose ($P_{Dose} < 0.05$) are presented in Fig 2.

### Biogenic amines

Serum concentrations of biogenic amines are presented in Table 2. The total concentration of serum biogenic amines was greater with choline at 6 x and 8 x NRC RA, as compared to control and 2 x NRC RA ($P_{Dose} < 0.001$). Specifically, trans-4-hydroxyproline and methionine sulfoxide were significantly affected by choline dose ($P_{Dose} = 0.046,$ and $0.020$). Although choline at 6 x NRC RA produced the highest serum concentration of both of these metabolites, there were no differences between choline doses when a Tukey's posthoc adjustment was applied. An effect of choline dose was observed for sarcosine ($P_{Dose} < 0.001$). Serum sarcosine concentrations were increased with choline at 4 x, 6 x and 8 x NRC RA, as compared to control and 2 x NRC RA. Similarly, serum trimethylamine N-oxide was higher with choline at 8 x NRC RA, as compared to control, 2 x and 4 x NRC RA, and at 4 x and 6 x NRC RA as compared to 2 x NRC RA and control ($P_{Dose} < 0.001$). Choline dose also affected putrescine ($P_{Dose} = 0.009$). Choline at 8 x NRC RA produced higher serum concentrations of putrescine when compared to the control treatment. A trend was noted for both serum carnosine and creatinine ($P_{Dose} = 0.080,$ and $0.083$, respectively). There were no significant changes noted in the serum concentrations of the other biogenic amines determined by DI-MS ($P_{Dose} > 0.05$), including: acetylornithine, asymmetric dimethylarginine, total dimethylarginine, alpha-amino adipic acid, histamine, kynurenine, serotonin, spermidine, spermine and tyramine.

### Amino acids, amino acid derivatives and ammonium compounds

Serum concentrations of amino acids, amino acid derivatives and ammonium compounds are presented in Table 3. Although total amino acid concentrations were not affected by choline

**Table 2. Mean serum concentrations (μM) of biogenic amines determined by DI-MS in overweight cats (n = 14) receiving control (no additional choline supplementation, 1.2 x NRC RA, 77 mg/kg BW$^{0.67}$), choline at 2 x NRC RA (126 mg/kg BW$^{0.67}$), 4 x NRC RA (252 mg/kg BW$^{0.67}$), 6 x NRC RA (378 mg/kg BW$^{0.67}$), and 8 x NRC RA (504 mg/kg BW$^{0.67}$), in a 5 x 5 Latin square design for 3-week periods.**

| Biogenic Amines | Choline Dose | | | | | $P_{Dose}$ |
| --- | --- | --- | --- | --- | --- | --- |
| | Control | 2 x NRC RA | 4 x NRC RA | 6 X NRC RA | 8 X NRC RA | |
| Acetylornithine | 109.27 ± 13.99 | 102.41 ± 13.88 | 104.15 ± 13.80 | 99.80 ± 13.80 | 95.02 ± 13.80 | 0.482 |
| Alpha amino adipic acid | 1.37 ± 0.14 | 1.46 ± 0.14 | 1.39 ± 0.14 | 1.48 ± 0.14 | 1.37 ± 0.14 | 0.867 |
| Asymmetric dimethylarginine | 1.10 ± 0.05 | 1.10 ± 0.04 | 1.10 ± 0.04 | 1.13 ± 0.04 | 1.16 ± 0.04 | 0.570 |
| Carnosine | 38.36 ± 5.46 | 38.05 ± 5.40 | 41.70 ± 5.35 | 45.45 ± 5.35 | 45.62 ± 5.35 | 0.080 |
| Creatinine | 132.81 ± 6.66 | 133.61 ± 6.44 | 136.63 ± 6.25 | 152.24 ± 6.25 | 140.72 ± 6.25 | 0.083 |
| Histamine | 0.086 ± 0.01 | 0.092 ± 0.01 | 0.092 ± 0.01 | 0.10 ± 0.01 | 0.10 ± 0.01 | 0.303 |
| Kynurenine | 9.47 ± 0.50 | 9.12 ± 0.49 | 9.08 ± 0.48 | 9.34 ± 0.48 | 9.13 ± 0.48 | 0.893 |
| Methionine sulfoxide | 5.33 ± 0.41 | 5.13 ± 0.40 | 5.91 ± 0.39 | 6.12 ± 0.39 | 5.18 ± 0.39 | **0.020** |
| Putrescine | 1.06 ± 0.09[b] | 1.16 ± 0.09[ab] | 1.25 ± 0.08[ab] | 1.35 ± 0.08[ab] | 1.42 ± 0.08[a] | **0.009** |
| Sarcosine | 5.55 ± 0.49[b] | 6.18 ± 0.47[b] | 8.04 ± 0.46[a] | 8.35 ± 0.46[a] | 8.32 ± 0.46[a] | **< 0.0001** |
| Serotonin | 28.50 ± 4.62 | 26.88 ± 4.49 | 29.77 ± 4.38 | 34.84 ± 4.38 | 35.09 ± 4.38 | 0.307 |
| Spermidine | 0.33 ± 0.03 | 0.34 ± 0.03 | 0.33 ± 0.03 | 0.36 ± 0.03 | 0.36 ± 0.03 | 0.659 |
| Spermine | 0.10 ± 0.01 | 0.11 ± 0.01 | 0.10 ± 0.01 | 0.11 ± 0.01 | 0.11 ± 0.01 | 0.437 |
| Total dimethylarginine | 4.53 ± 0.22 | 4.63 ± 0.22 | 4.61 ± 0.22 | 4.72 ± 0.22 | 4.78 ± 0.22 | 0.761 |
| Trans-4-hydroxyproline | 39.03 ± 2.29 | 39.70 ± 2.22 | 44.40 ± 2.16 | 44.51 ± 2.16 | 39.36 ± 2.16 | **0.046** |
| Trimethylamine n-oxide | 3.30 ± 6.52[c] | 9.56 ± 6.28[c] | 37.56 ± 6.06[b] | 59.79 ± 6.06[ab] | 77.42 ± 6.06[a] | **< 0.0001** |
| Tyramine | 0.085 ± 0.006 | 0.074 ± 0.006 | 0.079 ± 0.006 | 0.080 ± 0.006 | 0.083 ± 0.006 | 0.298 |
| Total biogenic amines | 382.54 ± 17.35[b] | 378.35 ± 17.27[b] | 425.36 ± 17.07[ab] | 470.06 ± 16.48[a] | 463.67 ± 17.05[a] | **< 0.0001** |

Values expressed as LSM ± SEM; Values in a row with superscripts without a common letter differ; P < 0.05, Repeated measures ANOVA with Tukey post-hoc test.

DI-MS = direct infusion mass spectrometry; BW = body weight; NRC = National Research Council; RA = Recommended Allowance

dose ($P_{Dose}$ = 0.179), significant changes were observed in numerous individual amino acids. Of the gluconeogenic amino acids, serum concentrations of arginine were greater with choline at 8 x NRC RA, as compared to control ($P_{Dose}$ = 0.039). Similarly, asparagine, proline and serine were significantly affected by choline dose ($P_{Dose}$ = 0.003, < 0.0001, and 0.001, respectively). The 4 x, 6 x, and 8 x NRC RA choline doses resulted in higher serum concentrations of all three of these aforementioned amino acids as compared to 2 x NRC RA and control. Serum methionine was greater at 4 x, 6 x and 8 x NRC RA, when compared to control ($P_{Dose}$ = 0.001). Although there was a significant effect of choline dose on serum histidine concentrations ($P_{Dose}$ = 0.014), no differences were observed between dose means when a Tukey's posthoc adjustment was applied. Of the ketogenic amino acids, only serum lysine increased with choline at 4 x and 8 x NRC, in comparison to control ($P_{Dose}$ = 0.006). Serum homocysteine concentrations were greatest with choline at 6 x NRC RA and lowest with choline at 8 x NRC RA ($P_{Dose}$ = 0.019). Choline dose did not affect the serum concentrations of the remaining amino acids analyzed by DI-MS ($P_{Dose}$ > 0.05), including: alanine, aspartic acid, citrulline, glutamic acid, glutamine, glycine, isoleucine, leucine, ornithine, phenylalanine, taurine, threonine, tyrosine, tryptophan, and valine.

When assessing the serum amino acid derivatives, only a significant effect of choline dose was observed for betaine ($P_{Dose}$ < 0.0001). Serum betaine concentrations were significantly greater with choline at 6 x and 8 x NRC RA, in comparison to 4 x, 2 x NRC RA, and control. Serum betaine was also greater with choline at 4 x NRC RA, as compared to 2 x NRC RA and control. No significant choline dose-related changes were observed in either serum methylhistidine or creatine ($P_{Dose}$ = 0.754, and 0.477, respectively). Additionally, serum concentrations of choline did not change dependent on choline dose ($P_{Dose}$ = 0.835).

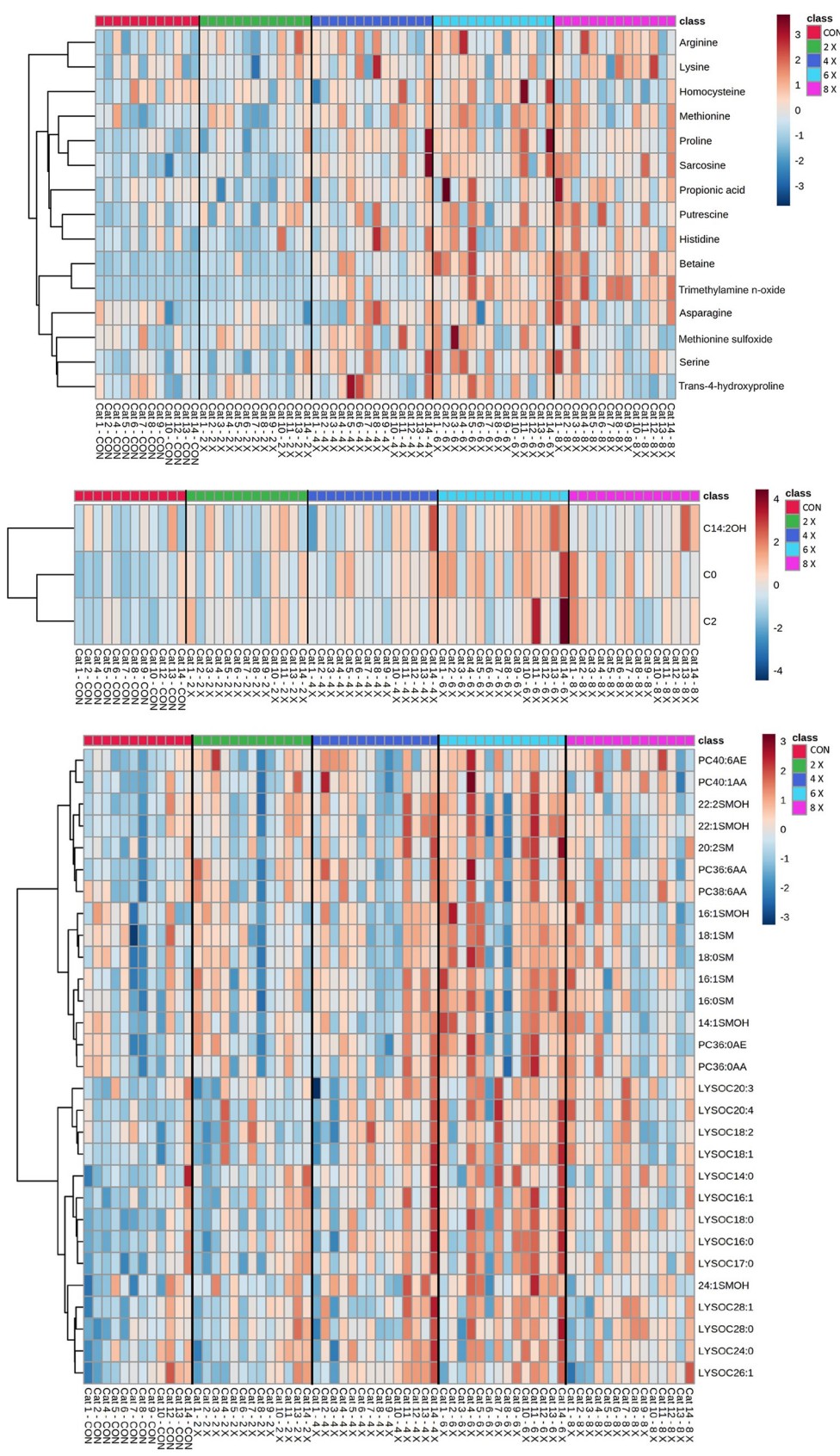

**Fig 2. Heatmap with euclidean distance and ward clustering of mean serum metabolites determined by DI-MS in overweight cats (n = 14) receiving control (no additional choline supplementation, 1.2 x NRC RA, 77 mg/kg BW0.67), choline at 2 x NRC RA (126 mg/kg BW0.67), 4 x NRC RA (252 mg/kg BW0.67), 6 x NRC RA (378 mg/kg BW0.67), and 8 x NRC RA (504 mg/kg BW0.67), in a 5 x 5 Latin square design for 3-week periods, with a significant effect of choline dose (P < 0.05).** Metabolites are separated into **a)** biogenic amines, amino acids, amino acid derivatives, ammonium compounds, organic sugars and acids; **b)** acylcarnitines; **c)** phosphatidylcholines, lysophosphatidylcholines, sphingomyelins and hydroxysphingomyelins.

**Table 3. Mean serum concentrations (µM) of amino acids, amino acid derivatives, and ammonium compounds determined by DI-MS in overweight cats (n = 14) receiving control (no additional choline supplementation, 1.2 x NRC RA, 77 mg/kg BW$^{0.67}$), choline at 2 x NRC RA (126 mg/kg BW$^{0.67}$), 4 x NRC RA (252 mg/kg BW$^{0.67}$), 6 x NRC RA (378 mg/kg BW$^{0.67}$), and 8 x NRC RA (504 mg/kg BW$^{0.67}$), in a 5 x 5 Latin square design for 3-week periods.**

| | Choline Dose | | | | | |
| --- | --- | --- | --- | --- | --- | --- |
| | **Control** | **2 x NRC RA** | **4 x NRC RA** | **6 X NRC RA** | **8 X NRC RA** | **P$_{Dose}$** |
| **Amino Acids** | | | | | | |
| Alanine | 443.89 ± 48.10 | 459.87 ± 48.54 | 477.19 ± 49.19 | 543.70 ± 56.04 | 512.11 ± 52.79 | 0.350 |
| Arginine | 162.25 ± 8.03[b] | 177.01 ± 7.84[ab] | 179.34 ± 7.68[ab] | 180.43 ± 7.68[ab] | 187.43 ± 7.68[a] | **0.039** |
| Asparagine | 47.15 ± 3.85[ab] | 45.30 ± 3.81[b] | 52.66 ± 3.77[a] | 53.55 ± 3.77[a] | 53.81 ± 3.77[a] | **0.003** |
| Aspartic acid | 62.61 ± 8.52 | 65.30 ± 8.34 | 69.08 ± 8.19 | 75.92 ± 8.19 | 75.56 ± 8.19 | 0.308 |
| Citrulline | 22.73 ± 1.59 | 23.61 ± 1.56 | 25.18 ± 1.53 | 25.51 ± 1.53 | 24.33 ± 1.53 | 0.266 |
| Glutamic acid | 93.92 ± 14.73 | 111.36 ± 14.64 | 105.75 ± 14.55 | 104.25 ± 14.55 | 95.86 ± 14.55 | 0.311 |
| Glutamine | 740.68 ± 37.34 | 694.80 ± 36.52 | 755.70 ± 35.80 | 696.03 ± 35.80 | 727.72 ± 35.80 | 0.295 |
| Glycine | 336.20 ± 14.64 | 348.06 ± 14.27 | 360.79 ± 13.94 | 368.74 ± 13.94 | 360.14 ± 13.94 | 0.222 |
| Histidine | 106.29 ± 3.33 | 107.62 ± 3.22 | 115.98 ± 3.11 | 117.20 ± 3.11 | 117.01 ± 3.11 | **0.014** |
| Homocysteine | 8.28 ± 0.31[ab] | 7.81 ± 0.29[ab] | 8.07 ± 0.28[ab] | 8.75 ± 0.28[a] | 7.52 ± 0.28[b] | **0.019** |
| Isoleucine | 55.18 ± 3.08 | 55.69 ± 3.02 | 56.22 ± 2.97 | 56.22 ± 2.97 | 57.13 ± 2.97 | 0.965 |
| Leucine | 113.00 ± 6.43 | 112.00 ± 6.29 | 114.90 ± 6.18 | 114.82 ± 6.18 | 116.98 ± 6.18 | 0.922 |
| Lysine | 152.50 ± 8.19[b] | 164.75 ± 8.01[ab] | 175.50 ± 7.85[a] | 170.68 ± 7.84[ab] | 181.56 ± 7.85[a] | **0.006** |
| Methionine | 43.45 ± 2.24[b] | 45.10 ± 2.20[bc] | 49.68 ± 2.17[ac] | 50.37 ± 2.17[a] | 49.75 ± 2.17[ac] | **0.001** |
| Ornithine | 36.34 ± 5.34 | 33.07 ± 5.19 | 38.41 ± 5.05 | 44.29 ± 5.05 | 39.70 ± 5.05 | 0.378 |
| Phenylalanine | 79.04 ± 4.17 | 83.00 ± 4.11 | 82.01 ± 4.06 | 84.37 ± 4.06 | 82.67 ± 4.06 | 0.527 |
| Proline | 138.03 ± 5.90[b] | 139.61 ± 5.75[b] | 157.52 ± 5.62[a] | 166.10 ± 5.62[a] | 157.69 ± 5.62[a] | **< 0.0001** |
| Serine | 137.41 ± 6.13[b] | 136.99 ± 5.94[b] | 159.11 ± 5.79[a] | 165.13 ± 5.78[a] | 157.41 ± 5.79[a] | **0.0001** |
| Taurine | 4.45 ± 0.19 | 4.61 ± 0.18 | 4.47 ± 0.17 | 4.65 ± 0.17 | 4.63 ± 0.17 | 0.864 |
| Threonine | 112.54 ± 5.32 | 114.61 ± 5.18 | 120.02 ± 5.06 | 120.32 ± 5.06 | 124.46 ± 5.06 | 0.203 |
| Tyrosine | 51.66 ± 3.23 | 49.95 ± 3.16 | 50.52 ± 3.09 | 51.80 ± 3.09 | 53.19 ± 3.09 | 0.837 |
| Tryptophan | 72.56 ± 2.72 | 73.75 ± 2.64 | 70.69 ± 2.57 | 71.31 ± 2.57 | 70.35 ± 2.57 | 0.759 |
| Valine | 152.13 ± 7.04 | 150.12 ± 6.92 | 157.83 ± 6.81 | 154.35 ± 6.81 | 153.76 ± 6.81 | 0.743 |
| Total amino acids | 3283.34 ± 133.91 | 3390.69 ± 130.02 | 3499.44 ± 126.55 | 3604.20 ± 126.55 | 3574.06 ± 126.55 | 0.179 |
| **Amino Acid Derivatives** | | | | | | |
| Betaine | 252.53 ± 58.92[c] | 358.69 ± 56.95[c] | 609.81 ± 55.23[b] | 851.65 ± 55.23[a] | 816.46 ± 55.23[a] | **< 0.0001** |
| Creatine | 9.55 ± 1.23 | 9.92 ± 1.19 | 9.81 ± 1.15 | 11.09 ± 1.15 | 11.93 ± 1.15 | 0.477 |
| Methylhistidine | 24.09 ± 1.33 | 23.24 ± 1.31 | 24.67 ± 1.29 | 23.82 ± 1.29 | 24.27 ± 1.29 | 0.754 |
| **Ammonium Compounds** | | | | | | |
| Choline | 18.98 ± 3.73 | 19.11 ± 3.66 | 22.56 ± 3.59 | 21.89 ± 3.59 | 20.77 ± 3.59 | 0.835 |

Values expressed as LSM ± SEM; Values in a row with superscripts without a common letter differ; P < 0.05, Repeated measures ANOVA with Tukey post-hoc test.

DI-MS = direct infusion mass spectrometry; BW = body weight; NRC = National Research Council; RA = Recommended Allowance.

## Acylcarnitines

Serum concentrations of acylcarnitines analyzed by DI-MS are presented in Table 4. The total concentration of serum acylcarnitines was significantly affected by choline dose ($P_{Dose} < 0.001$). Total serum acylcarnitines increased with choline at 4 x, 6 x and 8 x NRC RA as compared to control. Total acylcarnitines were also significantly greater with the 6 x NRC RA dose in comparison to 2 x NRC RA. The serum concentration of free acylcarnitines (C0) was similarly greater with choline intake at 4 x, 6 x and 8 x NRC RA, when compared to control ($P_{Dose} < 0.001$). Free acylcarnitines were also greater with 6 x NRC RA, in comparison to 2 x NRC RA.

For the short-chain acylcarnitines, the overall concentration of total short-chain acylcarnitines was greater with 6 x NRC RA, in comparison to control ($P_{Dose} = 0.017$). The only individual short-chain acylcarnitine affected by choline dose was C2 ($P_{Dose} = 0.010$), with choline at 6 x NRC RA resulting in increased serum concentrations of C2 as compared to control. Serum concentrations of C3:OH and C5:1 tended to change with choline dose ($P_{Dose} = 0.062$, and 0.051, respectively) and no differences were noted in the serum concentrations of the remaining short-chain acylcarnitines with choline dose ($P_{Dose} > 0.05$), including: C3, C3:1, C4, C4 OH, C4:1, C5, C5 OH, C5 DC, C5 MDC, and C5:1 DC.

Choline dose did not affect the total concentration of serum medium-chain acylcarnitines ($P_{Dose} = 0.268$). Additionally, there were no differences in the serum concentrations of any of the individual medium-chain acylcarnitines ($P_{Dose} > 0.05$), which included: C6, C6:1, C7:DC, C8, C9, C10, C10:1, C10:2, C12, C12:DC, and C12:1.

The total concentration of long-chain acylcarnitines did not change with choline dose ($P_{Dose} = 0.619$). Only C14:2 was affected by dose ($P_{Dose} = 0.029$), with choline at 6 X NRC RA increasing serum C14:2 concentrations as compared to control. There was a tendency for serum C18:2 ($P_{Dose} = 0.059$). No differences were observed with choline dose in the other long-chain acylcarnitines analyzed by DI-MS ($P_{Dose} > 0.05$), including: C14, C14:1, C14 OH, C14:2, C16, C16 OH, C16:1, C16:1 OH, C16:2, C16:2 OH, C18, C18:1, and C18:1 OH.

## Phosphatidylcholines

Serum concentrations of PC, both diacyl (PC aa) and acyl-alkyl (PC ae) analyzed by DI-MS are presented in Table 5. Overall, the total serum concentrations of both PC aa and PC ae were affected by choline dose ($P_{Dose} = 0.018$, and 0.012, respectively). Both total PC aa and total PC ae were greatest with choline at 6 x NRC RA, as compared to control. Of the PC aa metabolites assessed via DI-MS, there was an effect of choline dose on the concentrations of PC aa C36:0, PC aa C36:6, PC aa C38:6, and PC aa C40:1 ($P_{Dose} < 0.05$). Specifically, choline at 6 x NRC RA resulted in higher serum concentrations of PC aa C38:6 and PC aa C40:1, when compared to control ($P_{Dose} = 0.017$, and 0.023, respectively). Choline at 6 x NRC similarly resulted in greater serum concentrations of PC aa C36:0 in comparison to control, in addition to 2 x NRC RA ($P_{Dose} = 0.007$). Serum concentrations of PC aa C36:6 were increased with 4 x and 6 x NRC RA, as compared to control ($P_{Dose} < 0.0001$). Serum PC aa C36:6 with 6 x NRC RA choline was also significantly greater than 2 x and 8 x NRC RA. A tendency was observed for the four remaining PC aa metabolites; serum PC aa C32:2, PC aa C38:0, PC aa C40:2, and PC aa C40:6; to change with choline dose ($P_{Dose} = 0.079$, 0.063, 0.092, and 0.068, respectively). Both PC ae metabolites, PC ae C36:0 and PC ae C40:6, had increased serum concentrations with 6 x NRC RA choline when compared to control ($P_{Dose} = 0.012$, and 0.015, respectively).

## Lysophosphatidylcholines

Serum concentrations of LPC analyzed by DI-MS are presented in Table 6. The total concentration of serum LPC was greatest with choline at 6 x NRC RA, when compared to control and

**Table 4. Mean serum concentrations (μM) of acylcarnitines determined by DI-MS in overweight cats (n = 14) receiving control (no additional choline supplementation, 1.2 x NRC RA, 77 mg/kg BW$^{0.67}$), choline at 2 x NRC RA (126 mg/kg BW$^{0.67}$), 4 x NRC RA (252 mg/kg BW$^{0.67}$), 6 x NRC RA (378 mg/kg BW$^{0.67}$), and 8 x NRC RA (504 mg/kg BW$^{0.67}$), in a 5 x 5 Latin square design for 3-week periods.**

| | Choline Dose | | | | | |
| --- | --- | --- | --- | --- | --- | --- |
| | **Control** | **2 x NRC RA** | **4 x NRC RA** | **6 X NRC RA** | **8 X NRC RA** | **P$_{Dose}$** |
| Free acylcarnitines (C0) | 9.78 ± 0.58$^c$ | 11.20 ± 0.56$^{bc}$ | 12.13 ± 0.54$^{ab}$ | 13.77 ± 0.54$^a$ | 12.60 ± 0.54$^{ab}$ | **< 0.0001** |
| **Short Chain Acylcarnitines** | | | | | | |
| C2 | 0.74 ± 0.068$^b$ | 0.86 ± 0.066$^{ab}$ | 0.86 ± 0.064$^{ab}$ | 1.03 ± 0.064$^a$ | 0.94 ± 0.064$^{ab}$ | **0.010** |
| C3 | 0.097 ± 0.007 | 0.12 ± 0.009 | 0.11 ± 0.008 | 0.12 ± 0.009 | 0.11 ± 0.008 | 0.165 |
| C3:OH | 0.038 ± 0.002 | 0.043 ± 0.002 | 0.036 ± 0.002 | 0.038 ± 0.002 | 0.042 ± 0.002 | 0.062 |
| C3:1 | 0.032 ± 0.002 | 0.030 ± 0.002 | 0.028 ± 0.002 | 0.029 ± 0.002 | 0.032 ± 0.002 | 0.617 |
| C4 | 0.051 ± 0.005 | 0.064 ± 0.006 | 0.059 ± 0.006 | 0.062 ± 0.006 | 0.061 ± 0.006 | 0.232 |
| C4:OH | 0.014 ± 0.001 | 0.015 ± 0.001 | 0.014 ± 0.001 | 0.015 ± 0.001 | 0.013 ± 0.001 | 0.845 |
| C4:1 | 0.025 ± 0.001 | 0.024 ± 0.001 | 0.023 ± 0.001 | 0.023 ± 0.001 | 0.026 ± 0.001 | 0.387 |
| C5 | 0.055 ± 0.006 | 0.062 ± 0.006 | 0.064 ± 0.006 | 0.070 ± 0.006 | 0.067 ± 0.006 | 0.184 |
| C5:OH | 0.018 ± 0.001 | 0.018 ± 0.001 | 0.018 ± 0.001 | 0.017 ± 0.001 | 0.016 ± 0.001 | 0.625 |
| C5:DC | 0.013 ± 0.001 | 0.013 ± 0.001 | 0.011 ± 0.001 | 0.012 ± 0.001 | 0.012 ± 0.001 | 0.150 |
| C5:MDC | 0.017 ± 0.001 | 0.018 ± 0.001 | 0.017 ± 0.001 | 0.017 ± 0.001 | 0.018 ± 0.001 | 0.879 |
| C5:1 | 0.022 ± 0.001 | 0.019 ± 0.001 | 0.021 ± 0.001 | 0.022 ± 0.001 | 0.020 ± 0.001 | 0.051 |
| C5:1DC | 0.013 ± 0.001 | 0.011 ± 0.001 | 0.012 ± 0.001 | 0.011 ± 0.001 | 0.012 ± 0.001 | 0.109 |
| Total short chain acylcarnitines | 1.14 ± 0.084$^b$ | 1.31 ± 0.081$^{ab}$ | 1.28 ± 0.079$^{ab}$ | 1.47 ± 0.079$^a$ | 1.38 ± 0.079$^{ab}$ | **0.017** |
| **Medium Chain Acylcarnitines** | | | | | | |
| C6 | 0.045 ± 0.005 | 0.060 ± 0.006 | 0.050 ± 0.005 | 0.049 ± 0.005 | 0.055 ± 0.006 | 0.122 |
| C6:1 | 0.026 ± 0.002 | 0.028 ± 0.002 | 0.026 ± 0.002 | 0.029 ± 0.002 | 0.025 ± 0.002 | 0.182 |
| C7:DC | 0.016 ± 0.004 | 0.018 ± 0.005 | 0.015 ± 0.004 | 0.022 ± 0.006 | 0.018 ± 0.005 | 0.782 |
| C8 | 0.017 ± 0.002 | 0.021 ± 0.003 | 0.019 ± 0.002 | 0.017 ± 0.002 | 0.020 ± 0.002 | 0.397 |
| C9 | 0.016 ± 0.001 | 0.015 ± 0.001 | 0.015 ± 0.001 | 0.015 ± 0.001 | 0.015 ± 0.001 | 0.851 |
| C10 | 0.040 ± 0.006 | 0.046 ± 0.006 | 0.043 ± 0.006 | 0.039 ± 0.006 | 0.050 ± 0.006 | 0.648 |
| C10:1 | 0.062 ± 0.004 | 0.068 ± 0.004 | 0.064 ± 0.004 | 0.061 ± 0.004 | 0.065 ± 0.004 | 0.722 |
| C10:2 | 0.027 ± 0.002 | 0.028 ± 0.002 | 0.028 ± 0.002 | 0.028 ± 0.002 | 0.026 ± 0.002 | 0.845 |
| C12 | 0.030 ± 0.003 | 0.038 ± 0.004 | 0.032 ± 0.003 | 0.031 ± 0.003 | 0.036 ± 0.004 | 0.165 |
| C12:DC | 0.012 ± 0.001 | 0.012 ± 0.001 | 0.012 ± 0.001 | 0.012 ± 0.001 | 0.011 ± 0.001 | 0.719 |
| C12:1 | 0.043 ± 0.002 | 0.049 ± 0.002 | 0.044 ± 0.002 | 0.045 ± 0.002 | 0.048 ± 0.002 | 0.272 |
| Total medium chain acylcarnitines | 0.34 ± 0.023 | 0.41 ± 0.026 | 0.35 ± 0.022 | 0.36 ± 0.023 | 0.39 ± 0.024 | 0.268 |
| **Long Chain Acylcarnitines** | | | | | | |
| C14 | 0.025 ± 0.003 | 0.030 ± 0.003 | 0.028 ± 0.003 | 0.030 ± 0.003 | 0.031 ± 0.003 | 0.452 |
| C14:1 | 0.045 ± 0.004 | 0.055 ± 0.004 | 0.050 ± 0.004 | 0.047 ± 0.004 | 0.051 ± 0.004 | 0.249 |
| C14:1OH | 0.015 ± 0.001 | 0.015 ± 0.001 | 0.014 ± 0.001 | 0.015 ± 0.001 | 0.014 ± 0.001 | 0.544 |
| C14:2 | 0.013 ± 0.001 | 0.014 ± 0.001 | 0.012 ± 0.001 | 0.012 ± 0.001 | 0.014 ± 0.001 | 0.417 |
| C14:2 OH | 0.016 ± 0.001$^b$ | 0.016 ± 0.001$^{ab}$ | 0.017 ± 0.001$^{ab}$ | 0.018 ± 0.001$^a$ | 0.017 ± 0.001$^{ab}$ | **0.029** |
| C16 | 0.13 ± 0.001 | 0.14 ± 0.001 | 0.14 ± 0.001 | 0.15 ± 0.001 | 0.13 ± 0.001 | 0.729 |
| C16:OH | 0.015 ± 0.001 | 0.016 ± 0.001 | 0.015 ± 0.001 | 0.017 ± 0.001 | 0.015 ± 0.001 | 0.074 |
| C16:1 | 0.039 ± 0.002 | 0.043 ± 0.002 | 0.040 ± 0.002 | 0.042 ± 0.002 | 0.040 ± 0.002 | 0.308 |
| C16:1OH | 0.017 ± 0.001 | 0.016 ± 0.001 | 0.016 ± 0.001 | 0.017 ± 0.001 | 0.017 ± 0.001 | 0.854 |
| C16:2 | 0.012 ± 0.001 | 0.015 ± 0.001 | 0.013 ± 0.001 | 0.014 ± 0.001 | 0.014 ± 0.001 | 0.076 |
| C16:2 OH | 0.012 ± 0.001 | 0.011 ± 0.001 | 0.011 ± 0.001 | 0.012 ± 0.001 | 0.012 ± 0.001 | 0.496 |
| C18 | 0.096 ± 0.006 | 0.10 ± 0.006 | 0.10 ± 0.006 | 0.11 ± 0.006 | 0.10 ± 0.006 | 0.750 |
| C18:1 | 0.12 ± 0.008 | 0.13 ± 0.008 | 0.13 ± 0.008 | 0.13 ± 0.008 | 0.12 ± 0.008 | 0.615 |
| C18:1 OH | 0.020 ± 0.001 | 0.019 ± 0.001 | 0.019 ± 0.001 | 0.018 ± 0.001 | 0.018 ± 0.001 | 0.805 |
| C18:2 | 0.047 ± 0.003 | 0.051 ± 0.003 | 0.051 ± 0.003 | 0.055 ± 0.003 | 0.048 ± 0.003 | 0.059 |

*(Continued)*

**Table 4.** (Continued)

| | Choline Dose | | | | | |
|---|---|---|---|---|---|---|
| | **Control** | **2 x NRC RA** | **4 x NRC RA** | **6 X NRC RA** | **8 X NRC RA** | **P$_{Dose}$** |
| Total long chain acylcarnitines | 0.63 ± 0.035 | 0.68 ± 0.034 | 0.66 ± 0.033 | 0.69 ± 0.033 | 0.66 ± 0.033 | 0.619 |
| Total acylcarnitines | 11.90 ± 0.66[c] | 13.61 ± 0.64[bc] | 14.42 ± 0.62[ab] | 16.28 ± 0.62[a] | 15.05 ± 0.62[ab] | **<0.0001** |

Values expressed as LSM ± SEM; Values in a row with superscripts without a common letter differ; P < 0.05, Repeated measures ANOVA with Tukey post-hoc test.

DI-MS = direct infusion mass spectrometry; BW = body weight; NRC = National Research Council; RA = Recommended Allowance

2 x NRC RA (P$_{Dose}$ < 0.0001). The concentration of total LPC was also greater with choline at 4 x NRC RA, in comparison to control. With the exception of LPC C26:0 (P$_{Dose}$ = 0.131), individual LPC metabolites were all significantly affected by choline dose (P$_{Dose}$ < 0.05). Serum concentrations of both LPC C14:0 and LPC C17:0 were greater with choline at 4 x and 6 x NRC RA, as compared to control (P$_{Dose}$ = 0.005, and < 0.0001, respectively). Both LPC C16:0 and LPC C16:1 had higher serum concentrations with choline at 6 x NRC RA, when compared to control, 2 x and 8 x NRC RA (P$_{Dose}$ ≤ 0.0001, and 0.002, respectively). Additionally, the concentration of both these metabolites was greater with 4 x NRC RA, when compared to control. Choline at 6 x NRC RA resulted in higher serum concentrations of LPC C24:0, LPC C28:0, and LPC C28:1, in comparison to control (P$_{Dose}$ = 0.011, 0.017, and 0.018, respectively). Serum LPC C18:0 concentrations increased with 4 x, 6 x, and 8 x NRC, as compared to control (P$_{Dose}$ ≤ 0.0001). Choline at 6 x NRC RA also resulted in greater serum concentrations of LPC C18:0 as compared to 2 x NRC RA. Similarly, choline at an intake of 6 x NRC RA resulted in greater serum concentrations of LPC C18:1, when compared to 2 x NRC RA and control (P$_{Dose}$ = < 0.0001). Choline at 4 x NRC RA also increased LPC C18:1, as compared to control.

**Table 5. Mean serum concentrations (μM) of PC aa and PC ae determined by DI-MS in overweight cats (n = 14) receiving control (no additional choline supplementation, 1.2 x NRC RA, 77 mg/kg BW$^{0.67}$), choline at 2 x NRC RA (126 mg/kg BW$^{0.67}$), 4 x NRC RA (252 mg/kg BW$^{0.67}$), 6 x NRC RA (378 mg/kg BW$^{0.67}$), and 8 x NRC RA (504 mg/kg BW$^{0.67}$), in a 5 x 5 Latin square design for 3-week periods.**

| | Choline Dose | | | | | |
|---|---|---|---|---|---|---|
| | **Control** | **2 x NRC RA** | **4 x NRC RA** | **6 X NRC RA** | **8 X NRC RA** | **P$_{Dose}$** |
| **PC aa** | | | | | | |
| PC aa C32:2 | 2.05 ± 0.09 | 2.11 ± 0.09 | 2.14 ± 0.09 | 2.28 ± 0.09 | 2.13 ± 0.09 | 0.079 |
| PC aa C36:0 | 27.28 ± 1.51[b] | 27.92 ± 1.48[b] | 29.38 ± 1.45[ab] | 32.27 ± 1.45[a] | 29.42 ± 1.45[ab] | **0.007** |
| PC aa C36:6 | 0.79 ± 0.06[c] | 0.89 ± 0.06[bc] | 0.95 ± 0.06[ab] | 1.03 ± 0.06[a] | 0.89 ± 0.06[bc] | **< 0.0001** |
| PC aa C38:0 | 3.39 ± 0.23 | 3.60 ± 0.22 | 3.71 ± 0.22 | 3.93 ± 0.22 | 3.73 ± 0.22 | 0.063 |
| PC aa C38:6 | 55.49 ± 3.54[b] | 58.41 ± 3.50[ab] | 60.20 ± 3.46[ab] | 63.90 ± 3.46[a] | 60.78 ± 3.46[ab] | **0.017** |
| PC aa C40:1 | 0.76 ± 0.05[b] | 0.82 ± 0.05[ab] | 0.84 ± 0.05[ab] | 0.92 ± 0.05[a] | 0.83 ± 0.05[ab] | **0.023** |
| PC aa C40:2 | 1.69 ± 0.21 | 1.87 ± 0.21 | 1.97 ± 0.20 | 2.12 ± 0.20 | 1.96 ± 0.20 | 0.092 |
| PC aa C40:6 | 49.42 ± 4.26 | 52.47 ± 4.23 | 52.40 ± 4.20 | 54.88 ± 4.20 | 55.91 ± 4.20 | 0.068 |
| Total PC aa | 140.72 ± 8.87[b] | 148.06 ± 8.77[ab] | 151.65 ± 8.68[ab] | 161.41 ± 8.68[a] | 155.74 ± 8.68[ab] | **0.018** |
| **PC ae** | | | | | | |
| PC ae C36:0 | 2.76 ± 0.11[b] | 2.89 ± 0.11[ab] | 2.97 ± 0.11[ab] | 3.16 ± 0.11[a] | 2.88 ± 0.11[ab] | **0.012** |
| PC ae C40:6 | 3.10 ± 0.20[b] | 3.26 ± 0.20[ab] | 3.34 ± 0.20[ab] | 3.55 ± 0.20[a] | 3.43 ± 0.20[ab] | **0.015** |
| Total PC ae | 5.85 ± 0.28[b] | 6.16 ± 0.27[ab] | 6.31 ± 0.27[ab] | 6.72 ± 0.27[a] | 6.31 ± 0.27[ab] | **0.012** |

Values expressed as LSM ± SEM; Values in a row with superscripts without a common letter differ; P < 0.05, Repeated measures ANOVA with Tukey post-hoc test.

PC = phosphatidylcholine; aa = diacyl; ae = acyl-alkyl; DI-MS = direct infusion mass spectrometry; BW = body weight; NRC = National Research Council;

RA = Recommended Allowance

**Table 6. Mean serum concentrations (μM) of LPC determined by DI-MS in overweight cats (n = 14) receiving control (no additional choline supplementation, 1.2 x NRC RA, 77 mg/kg BW$^{0.67}$), choline at 2 x NRC RA (126 mg/kg BW$^{0.67}$), 4 x NRC RA (252 mg/kg BW$^{0.67}$), 6 x NRC RA (378 mg/kg BW$^{0.67}$), and 8 x NRC RA (504 mg/kg BW$^{0.67}$), in a 5 x 5 Latin square design for 3-week periods.**

| LPC | Choline Dose | | | | | $P_{Dose}$ |
|---|---|---|---|---|---|---|
|  | Control | 2 x NRC RA | 4 x NRC RA | 6 X NRC RA | 8 X NRC RA |  |
| LPC C14:0 | 0.71 ± 0.05[b] | 0.76 ± 0.05[ab] | 0.84 ± 0.05[a] | 0.86 ± 0.05[a] | 0.79 ± 0.05[ab] | **0.005** |
| LPC C16:0 | 57.51 ± 2.87[b] | 61.62 ± 2.80[bc] | 65.95 ± 2.74[ac] | 71.66 ± 2.74[a] | 62.97 ± 2.74[bc] | **< 0.0001** |
| LPC C16:1 | 1.58 ± 0.12[c] | 1.71 ± 0.11[bc] | 1.86 ± 0.11[ab] | 2.05 ± 0.11[a] | 1.75 ± 0.11[bc] | **0.0002** |
| LPC C17:0 | 1.30 ± 0.07[b] | 1.38 ± 0.07[ab] | 1.51 ± 0.07[a] | 1.62 ± 0.07[a] | 1.47 ± 0.07[ab] | **< 0.0001** |
| LPC C18:0 | 64.93 ± 3.35[c] | 71.47 ± 3.26[bc] | 76.64 ± 3.18[ab] | 85.90 ± 3.18[a] | 77.86 ± 3.18[ab] | **< 0.0001** |
| LPC C18:1 | 24.95 ± 1.63[c] | 26.35 ± 1.60[bc] | 29.71 ± 1.57[ab] | 32.64 ± 1.57[a] | 28.96 ± 1.57[abc] | **< 0.0001** |
| LPC C18:2 | 44.22 ± 3.89[b] | 45.51 ± 3.83[b] | 51.98 ± 3.77[ab] | 55.41 ± 3.77[a] | 47.81 ± 3.77[ab] | **0.003** |
| LPC C20:3 | 2.16 ± 0.17 | 2.16 ± 0.17 | 2.40 ± 0.16 | 2.70 ± 0.16 | 2.57 ± 0.16 | **0.035** |
| LPC C20:4 | 8.09 ± 0.52[c] | 8.63 ± 0.51[bc] | 9.40 ± 0.50[abc] | 10.57 ± 0.50[a] | 9.73 ± 0.50[ab] | **0.0002** |
| LPC C24:0 | 0.27 ± 0.02[b] | 0.28 ± 0.02[ab] | 0.32 ± 0.02[ab] | 0.33 ± 0.02[a] | 0.29 ± 0.02[ab] | **0.011** |
| LPC C26:0 | 0.13 ± 0.02 | 0.15 ± 0.02 | 0.15 ± 0.01 | 0.17 ± 0.01 | 0.16 ± 0.01 | 0.131 |
| LPC C26:1 | 0.09 ± 0.01 | 0.09 ± 0.01 | 0.11 ± 0.01 | 0.11 ± 0.01 | 0.10 ± 0.01 | **0.044** |
| LPC C28:0 | 0.18 ± 0.02[b] | 0.21 ± 0.02[ab] | 0.22 ± 0.02[ab] | 0.25 ± 0.02[a] | 0.22 ± 0.02[ab] | **0.017** |
| LPC C28:1 | 0.19 ± 0.02[b] | 0.22 ± 0.02[ab] | 0.24 ± 0.02[ab] | 0.27 ± 0.02[a] | 0.24 ± 0.02[ab] | **0.018** |
| Total LPC | 207.22 ± 10.66[c] | 221.28 ± 10.38[bc] | 241.39 ± 10.13[ab] | 263.80 ± 10.13[a] | 234.58 ± 10.13[abc] | **< 0.0001** |

Values expressed as LSM ± SEM; Values in a row with superscripts without a common letter differ; P < 0.05, Repeated measures ANOVA with Tukey post-hoc test.

LPC = lysophosphatidylcholine; DI-MS = direct infusion mass spectrometry; BW = body weight; NRC = National Research Council; RA = Recommended Allowance.

Both serum LPC C18:2 and LPC C20:4 were elevated by the choline at 6 x NRC RA, when compared to 2 x NRC RA and control ($P_{Dose}$ = 0.003, and 0.002, respectively). Choline at 8 x NRC RA also increased LPC C20:4 in comparison to control. Although there was an effect of choline dose on both serum LPC C20:3 and LPC C26:1 ($P_{Dose}$ = 0.035, and 0.044, respectively), these differences were no longer significant when a Tukey's posthoc adjustment was applied.

## Sphingomyelines and hydroxysphingomyelines

Serum concentrations of SM and HSM analyzed by DI-MS are presented in Table 7. The serum concentration of total HSM was greatest with choline at 6 x NRC RA, as compared to control, 2 x and 8 x NRC RA ($P_{Dose}$ = 0.002). All individual HSM metabolites analyzed were affected by choline dose ($P_{Dose}$ < 0.05). Both HSM C14:1 and C22:1 had higher serum concentrations with choline at 6 x NRC RA, in comparison to control, 2 x and 8 x NRC RA ($P_{Dose}$ = 0.001, and 0.001, respectively). Similarly, 6 x NRC RA resulted in increased HSM C22:2 and HSM C24:1, when compared to control and 8 x NRC RA ($P_{Dose}$ = 0.008, and 0.008, respectively). Serum concentrations of HSM C16:1 were higher with 6 x NRC RA choline compared to the control treatment ($P_{Dose}$ = 0.008).

Similar to total HSM, total SM was also greatest with choline at 6 x NRC RA, when compared to control, 2 x and 8 x NRC RA ($P_{Dose}$ = 0.002). Similarly, all SM metabolites presented significant differences in their mean serum concentrations with choline dose ($P_{Dose}$ < 0.05). Serum concentrations of SM C16:0, SM 16:1 and SM C20:2 were highest with choline at 6 x NRC RA, in comparison to control, 2 x and 8 x NRC RA ($P_{Dose}$ = 0.001, 0.001, and 0.0001, respectively). The 6 x NRC RA choline dose also resulted in greater serum concentrations of SM C18:0 as compared to control and 2 x NRC RA ($P_{Dose}$ = 0.007), in addition to greater concentrations of SM 18:1 as compared to control ($P_{Dose}$ = 0.013).

**Table 7. Mean serum concentrations (μM) of HSM and SM determined by DI-MS in overweight cats (n = 14) receiving control (no additional choline supplementation, 1.2 x NRC RA, 77 mg/kg BW$^{0.67}$), choline at 2 x NRC RA (126 mg/kg BW$^{0.67}$), 4 x NRC RA (252 mg/kg BW$^{0.67}$), 6 x NRC RA (378 mg/kg BW$^{0.67}$), and 8 x NRC RA (504 mg/kg BW$^{0.67}$), in a 5 x 5 Latin square design for 3-week periods.**

| | Choline Dose | | | | | |
| --- | --- | --- | --- | --- | --- | --- |
| | Control | 2 x NRC RA | 4 x NRC RA | 6 X NRC RA | 8 X NRC RA | P$_{Dose}$ |
| **HSM** | | | | | | |
| HSM C14:1 | 19.94 ± 0.98[b] | 20.96 ± 0.97[b] | 21.49 ± 0.95[ab] | 23.30 ± 0.95[a] | 20.63 ± 0.95[b] | **0.001** |
| HSM C16:1 | 7.47 ± 0.31[b] | 7.74 ± 0.31[ab] | 7.86 ± 0.30[ab] | 8.53 ± 0.30[a] | 7.76 ± 0.30[ab] | **0.008** |
| HSM C22:1 | 35.14 ± 1.61[b] | 36.46 ± 1.58[b] | 38.30 ± 1.55[ab] | 41.11 ± 1.55[a] | 36.39 ± 1.55[b] | **0.001** |
| HSM C22:2 | 14.92 ± 0.67[b] | 15.59 ± 0.66[ab] | 16.15 ± 0.64[ab] | 17.13 ± 0.64[a] | 15.27 ± 0.64[b] | **0.008** |
| HSM C24:1 | 5.88 ± 0.26[b] | 6.20 ± 0.25[ab] | 6.38 ± 0.25[ab] | 6.75 ± 0.25[a] | 5.99 ± 0.25[b] | **0.008** |
| Total HSM | 83.56 ± 3.29[b] | 86.76 ± 3.21[b] | 90.05 ± 3.14[ab] | 96.74 ± 3.14[a] | 86.13 ± 3.14[b] | **0.002** |
| **SM** | | | | | | |
| SM C16:0 | 226.93 ± 7.81[b] | 236.57 ± 7.58[b] | 243.39 ± 7.39[ab] | 262.82 ± 7.39[a] | 235.45 ± 7.39[b] | **0.001** |
| SM C16:1 | 10.82 ± 0.43[b] | 11.48 ± 0.42[b] | 11.61 ± 0.42[ab] | 12.66 ± 0.42[a] | 11.47 ± 0.42[b] | **0.001** |
| SM C18:0 | 47.72 ±1.80[b] | 49.41 ±1.75[b] | 49.81 ±1.72[ab] | 54.31 ±1.72[a] | 50.27 ±1.72[ab] | **0.007** |
| SM C18:1 | 10.07 ± 0.39[b] | 10.34 ± 0.39[ab] | 10.48 ± 0.38[ab] | 11.33 ± 0.38[a] | 10.41 ± 0.38[ab] | **0.013** |
| SM C20:2 | 0.81 ± 0.06[b] | 0.84 ± 0.05[b] | 0.92 ± 0.05[ab] | 1.02 ± 0.05[a] | 0.85 ± 0.05[b] | **0.0001** |
| Total SM | 297.02 ± 9.90[b] | 308.50 ± 9.60[b] | 316.07 ± 9.35[ab] | 341.73 ± 9.35[a] | 308.70 ± 0.35[b] | **0.002** |

Values expressed as LSM ± SEM; Values in a row with superscripts without a common letter differ; P < 0.05, Repeated measures ANOVA with Tukey post-hoc test.

HSM = Hydroxysphingomyelines; SM = sphingomyelines; DI-MS = direct infusion mass spectrometry; BW = body weight; NRC = National Research Council;

RA = Recommended Allowance

## Organic acids and sugars

Serum concentrations of organic sugars and acids are presented in Table 8. Of these metabolites, there was a significant effect of dose only on serum propionic acid (P$_{Dose}$ = 0.036). Concentrations of serum propionic acid were greater with choline at 6 x NRC RA, as compared to 2 x NRC RA. Additionally, a tendency was shown for serum hippuric acid (P$_{Dose}$ = 0.062). Choline dose did not affect the remaining serum organic acids and sugars determined by DI-MS (P$_{Dose}$ > 0.05), including: lactic acid, beta-hydroxybutyric acid, alpha-ketoglutaric acid, citric acid, butyric acid, 3-hydroxyphenyl-hydracrylic acid, succinic acid, fumaric acid, pyruvic acid, isobutyric acid, methylmalonic acid, indole acetic acid, uric acid, 5-hydroxyindole acetic acid, and glucose.

## Discussion

To the authors' knowledge, the present study is the first to investigate the fasting serum metabolomic profile of healthy male adult cats consuming varying doses of dietary choline. As previously described in growing kittens and other mammals [48, 57, 58], there is a clear relationship between dietary choline and one-carbon metabolism in adult cats. Concentrations of serum choline did not change with choline intake in the present study. This finding aligns with previous observations in growing kittens and mink [48, 59]. Because choline appears to be quickly metabolized upon absorption, circulating serum or plasma choline concentrations are not recommended as indicators of choline status in an animal.

Although serum choline concentrations did not change, serum betaine concentrations increased with dietary choline intake at 4, 6 and 8 x NRC RA, as compared to the control treatment (Fig 3). This was an expected finding as choline is irreversibly oxidized to betaine in a two-step reaction [60, 61]. Serum betaine appears to be a more representative indicator of

**Table 8. Mean serum concentrations (µM) of organic acids and sugars determined by DI-MS in overweight cats (n = 14) receiving control (no additional choline supplementation, 1.2 x NRC RA, 77 mg/kg BW$^{0.67}$), choline at 2 x NRC RA (126 mg/kg BW$^{0.67}$), 4 x NRC RA (252 mg/kg BW$^{0.67}$), 6 x NRC RA (378 mg/kg BW$^{0.67}$), and 8 x NRC RA (504 mg/kg BW$^{0.67}$), in a 5 x 5 Latin square design for 3-week periods.**

| Organic Acids & Sugars | Choline Dose | | | | | P$_{Dose}$ |
|---|---|---|---|---|---|---|
| | Control | 2 x NRC RA | 4 x NRC RA | 6 X NRC RA | 8 X NRC RA | |
| 3-Hydroxyphenyl-hydracrylic acid | 0.01 ± 0.0003 | 0.01 ± 0.0003 | 0.01 ± 0.0003 | 0.01 ± 0.0003 | 0.01 ± 0.0003 | 0.446 |
| 5-Hydroxyindole acetic acid | 0.060 ± 0.004 | 0.062 ± 0.004 | 0.061 ± 0.004 | 0.063 ± 0.004 | 0.063 ± 0.004 | 0.884 |
| Alpha-ketoglutaric acid | 11.88 ± 1.98 | 9.27 ± 1.97 | 11.06 ± 1.95 | 10.97 ± 1.95 | 11.57 ± 1.95 | 0.222 |
| Beta-hydroxybutyric acid | 26.83 ± 2.43 | 25.85 ± 2.38 | 26.97 ± 2.34 | 23.38 ± 2.34 | 22.63 ± 2.34 | 0.158 |
| Butyric acid | 0.30 ± 0.04 | 0.24 ± 0.03 | 0.32 ± 0.03 | 0.33 ± 0.03 | 0.31 ± 0.03 | 0.311 |
| Citric acid | 217.93 ± 12.21 | 202.61 ± 11.89 | 215.90 ± 11.62 | 217.11 ± 11.62 | 216.62 ± 11.62 | 0.729 |
| Fumaric acid | 0.79 ± 0.15 | 0.75 ± 0.15 | 0.75 ± 0.14 | 0.82 ± 0.16 | 0.93 ± 0.18 | 0.664 |
| Glucose | 10483.00 ± 932.69 | 9337.70 ± 910.67 | 9432.36 ± 891.08 | 9836.88 ± 891.06 | 9963.26 ± 891.08 | 0.739 |
| Hippuric Acid | 0.84 ± 0.29 | 1.05 ± 0.29 | 0.79 ± 0.29 | 1.46 ± 0.29 | 1.26 ± 0.29 | 0.062 |
| Indole acetic acid | 0.67 ± 0.07 | 0.57 ± 0.06 | 0.61 ± 0.06 | 0.58 ± 0.06 | 0.61 ± 0.06 | 0.657 |
| Isobutyric acid | 4.20 ± 0.63 | 4.16 ± 0.61 | 4.80 ± 0.69 | 4.90 ± 0.71 | 5.48 ± 0.79 | 0.275 |
| Lactic acid | 1260.85 ± 143.31 | 1378.19 ± 141.55 | 1160.13 ± 139.99 | 1259.58 ± 139.99 | 1347.10 ± 139.99 | 0.194 |
| Methylmalonic acid | 0.17 ± 0.02 | 0.18 ± 0.01 | 0.16 ± 0.01 | 0.17 ± 0.01 | 0.17 ± 0.01 | 0.782 |
| Propionic acid | 1.68 ± 0.20$^{ab}$ | 1.41 ± 0.19$^{b}$ | 1.89 ± 0.19$^{ab}$ | 2.03 ± 0.19$^{a}$ | 2.14 ± 0.19$^{ab}$ | **0.036** |
| Pyruvic acid | 15.26 ± 3.91 | 11.24 ± 3.78 | 12.97 ± 3.91 | 11.46 ± 3.78 | 10.78 ± 3.65 | 0.860 |
| Succinic acid | 1.25 ± 0.18 | 1.36 ± 0.17 | 1.50 ± 0.17 | 1.37 ± 0.17 | 1.35 ± 0.17 | 0.634 |
| Uric acid | 9.15 ± 0.76 | 8.80 ± 0.74 | 9.39 ± 0.72 | 9.10 ± 0.72 | 9.56 ± 0.72 | 0.884 |

Values expressed as LSM ± SEM; Values in a row with superscripts without a common letter differ; P < 0.05, Repeated measures ANOVA with Tukey post-hoc test.

DI-MS = direct infusion mass spectrometry; BW = body weight; NRC = National Research Council; RA = Recommended Allowance.

choline intake, as compared to serum choline. Betaine has an important role within one-carbon metabolism as a methyl group donor for the methylation of homocysteine to produce methionine through the re-methylation pathway [62]. Upon donating one of its methyl groups to homocysteine to yield methionine, betaine is converted to DMG. Sarcosine is formed from DMG when DMG releases one of its methyl groups [63]. It is therefore unsurprising that increased dietary choline intake at 6 and 8 x NRC RA not only increased serum betaine concentrations but also increased serum DMG and sarcosine (Fig 3). Sarcosine results in the production of glycine when its final methyl group is released [64]. Additionally, within the folate cycle, serine hydroxymethyltransferase (SHMT) facilitates the conversion of serine to glycine [65]. This reaction also results in the production of 5,10-methylene tetrahydrofolate (THF), which is later converted to 5-methyl THF by methylenetetrahydrofolate reductase (MTHFR) to re-methylate homocysteine (Fig 3) [66]. Thus, the increase in serum serine, without an increase in serum glycine, may suggest that the folate cycle was spared in favour of transmethylation by betaine. Growing kittens had similar increases in serum betaine, DMG, sarcosine and serine, without the increase in glycine, with additional choline supplementation [48].

As expected, the increased serum betaine concentrations led to increased serum methionine concentrations with choline doses of 6 and 8 x NRC RA (Fig 3). This was presumably due to the increase in re-methylation of homocysteine to methionine by betaine-homocysteine methyltransferase (BHMT) [62]. However, reduced demand for methionine for methylation reactions through SAMe may have also resulted in greater serum concentrations of methionine. A similar increase in circulating methionine was previously observed in obese adult cats and growing kittens receiving additional dietary choline supplementation [46, 48]. Although choline at 6 and 8 x NRC RA increased serum methionine in the present study, similar changes

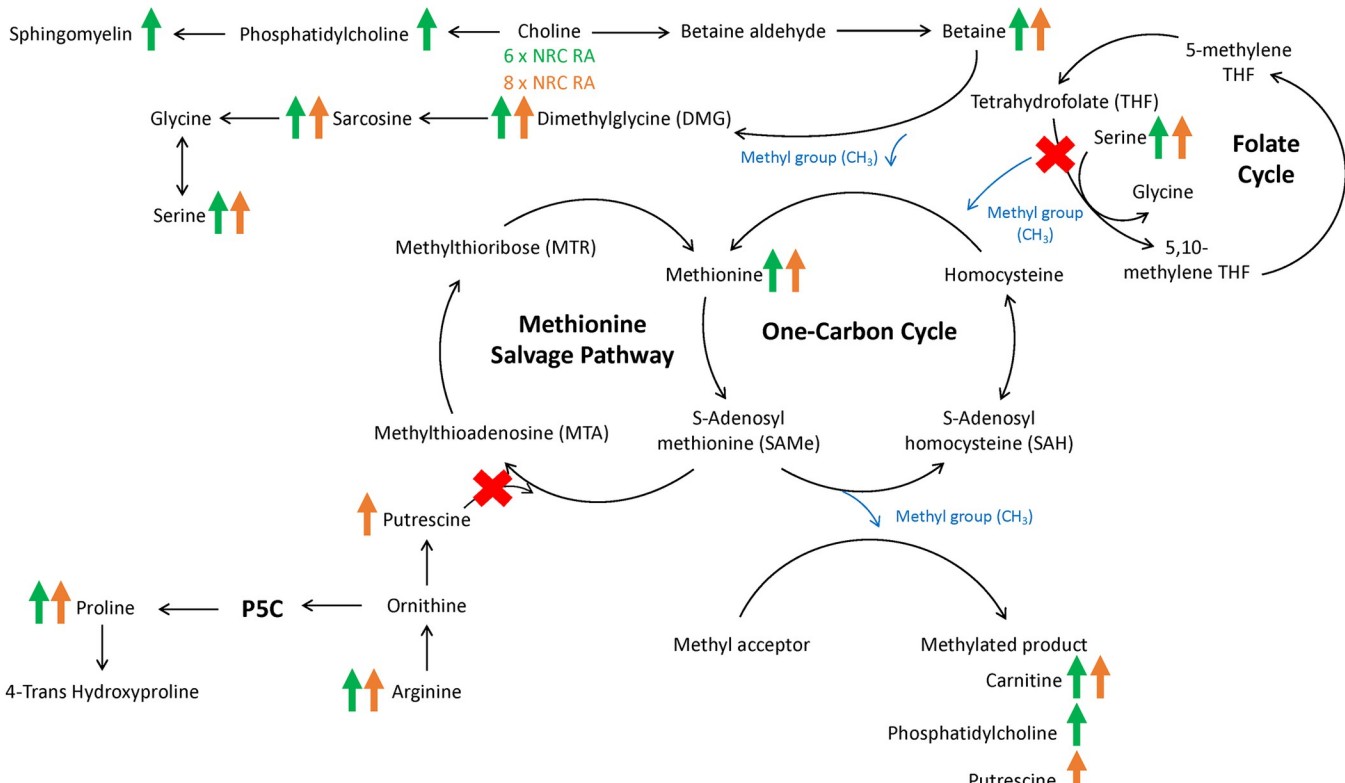

**Fig 3. Changes in mean serum metabolite concentrations in overweight male adult cats (n = 14) resulting from the consumption of dietary choline at 6 x NRC RA (378 mg/kg BW0.67) or 8 x NRC RA (504 mg/kg BW0.67) for 3-week periods.** Green and orange arrows represent increased serum metabolite concentrations with 6 x NRC RA or 8 x NRC RA, respectively. A red "X" represents a hypothesized decreased activity of said pathway.

were not observed for serum homocysteine. Of all the treatments provided, choline at 6 x NRC RA resulted in the highest concentration of homocysteine, while the 8 x NRC RA dose resulted in the lowest. It is unclear why these two doses resulted in such opposite differences in homocysteine, despite both causing similar increases in methionine. In dogs, additional methionine supplementation with a complete and balanced diet similarly increased plasma homocysteine concentrations, although plasma methionine concentrations remained unchanged [67]. Although there is no research investigating homocysteine concentrations with the supplementation of methyl donors in cats, serum homocysteine concentrations from both the 6 and 8 x NRC RA doses were similar to concentrations previously published in healthy cats [68, 69]. It is therefore unlikely that the higher homocysteine concentration with 6 x NRC RA would have any biological significance, or result in any clinical outcomes. In dogs and humans, hyperhomocysteinemia is associated with renal and cardiac diseases [70, 71]. There is no evidence to support the association between elevated concentrations of homocysteine and cardiac disease in cats [72]. Although there is limited published evidence suggesting that cats with chronic renal disease have higher concentrations of circulating homocysteine [73], the homocysteine concentrations of the cats with chronic renal disease and the healthy control cats in the aforementioned study were both higher than the concentrations presented herein with 6 x NRC RA ($41.68 \pm 9.97$ μM, $13.03 \pm 2.81$ μM, and $8.75 \pm 0.28$ μM, respectively).

The enhanced availability of methionine with increased dietary choline may have boosted the production and availability of universal methyl donor SAMe. However, this cannot be concluded with certainty as concentrations of SAMe were not assessed in the present study. As

mentioned previously, methylation facilitated by SAMe allows for the synthesis of a large number of metabolites, including carnitine (C0) [74]. In the present study, choline at 4, 6 and 8 x NRC RA increased serum concentrations of free carnitine (Fig 3). This finding was also observed in obese adult cats and growing kittens receiving supplemental dietary choline [46, 48]. An increase in carnitine concentrations suggests improved facilitation for fatty acid oxidation [75]. Acylcarnitines are the product of fatty acids entering the mitochondria for β-oxidation, and increased concentrations are suggestive of incomplete fatty acid oxidation [76, 77]. Increased acylcarnitine concentrations have also been associated with insulin resistance and obesity in humans and rodent models [77–79]. The supplementation of choline to mice upregulated AMPK and decreased acylcarnitine concentrations [80]. The activation of AMPK results in the upregulation of fatty acid oxidation and lipolysis, and the downregulation of lipogenesis [80, 81]. However, in the present study serum concentrations of C2, C14:2 OH, total short-chain and total combined acylcarnitines increased with choline supplementation at 6 x NRC RA, as compared to control. The findings of the present study also do not align with previous research in obese adult cats consuming choline at five times NRC RA [46]. The cats in that study had no changes in their plasma acylcarnitine concentrations and decreased acylcarnitine to free carnitine ratios with treatment, suggesting enhanced fatty acid utilization. In growing kittens, acetylcarnitine (C2) concentrations similarly increased with choline treatment after 12 weeks [48]. However, there were no increases in C14:2 OH, short-chain or total acylcarnitine concentrations in the kittens. Instead, the serum concentration of total medium-chain acylcarnitines decreased in the aforementioned kittens with choline treatment. As C2 is derived from acetyl-CoA, increased concentrations of C2 may be indicative of increased production of acetyl-CoA from β-oxidation that exceeds the capacity of the Kreb's cycle [82–84]. Additionally, C2 together with C0 (free carnitine) is one of the main circulating forms of L-carnitine. Therefore, the increase in C2 may also be indicative of increased L-carnitine biosynthesis through the increased availability of SAMe [74]. This hypothesis is supported by increases in plasma C2 in humans following the supplementation of dietary L-carnitine [85]. The increase in short-chain acylcarnitines with 6 x NRC RA in the present study was likely a reflection of the increase in C2. Increases in C14:2 OH have been reported in mice in correlation with increased plasma lipid concentrations [86]. Therefore, C14:2 OH may be reflective of dyslipidemia and/or hyperlipidemia [86, 87]. However, the pathways through which C14:2 OH acts and the clinical significance of increased C14:2OH in cats remains unclear and requires further elucidation.

Serum isopropanol was lowest with the 6 x NRC RA choline treatment and highest with the control treatment. Increased circulating isopropanol concentrations have previously been associated with increases in ketone bodies in livestock species [88, 89]. Endogenous isopropanol can be synthesized from ketone bodies (acetone and acetoacetate) produced by fatty acid oxidation [90]. Therefore, the decrease in isopropanol with choline at 6 x NRC RA observed herein may be due to acetyl-CoA being preferentially condensed with oxaloacetate for entrance into the Kreb's cycle, as opposed to ketone body formation. Alternatively, amino acid catabolism may have been reduced. Serum propionic acid increased with choline at 8 x NRC RA. Propionic acid can be produced by intestinal microbial fermentation of Kreb's cycle intermediates, the catabolism of certain amino acids (including methionine), or via biosynthesis from acetyl-CoA [91]. In the present study, it is unclear through which pathway dietary choline influenced the increase in propionic acid concentrations. The clinical considerations for changes in the serum concentrations of isopropanol and propionic acid in cats are unknown.

Additionally, SAMe is responsible for donating methyl groups for the biosynthesis of polyamines such as putrescine [92]. The increase in serum putrescine with choline at 8 x NRC RA may be reflective of the increased availability of SAMe with increased one-carbon metabolism

(Fig 3). However, putrescine also participates in the methionine salvage pathway [93]. As previously mentioned, the increased availability of serum betaine for the re-methylation of homocysteine to methionine likely caused other pathways responsible for the production of methionine, including the folate cycle and the methionine salvage pathway, to be spared. Arginine is converted to ornithine, leading to the production of putrescine. In the present study, we observed that concentrations of both arginine and putrescine increased with higher choline intake, which may suggest decreased activity of the methionine salvage pathway. In addition to producing putrescine, ornithine can also participate in the P5C pathway, leading to the synthesis of proline and thus 4-trans hydroxyproline [94]. Proline increased with choline intake at 4, 6 and 8 x NRC RA and 4-trans hydroxyproline increased with 6 x NRC RA, further suggesting a decreased demand for putrescine for the methionine salvage pathway (Fig 3). It is also important to note that the conversion of arginine to ornithine is dependent on arginase 1 (ARG1) within the urea cycle. As a result, the observed increases in ornithine-derived metabolites may be indicative of increased urea cycle activity with choline supplementation. However, this seems unlikely due to the lower concentrations of serum blood urea nitrogen with choline at 6 and 8 x NRC RA in these cats [47].

Apart from methionine, arginine and proline, choline supplementation at 6 and 8 x NRC RA increased the concentrations of several other amino acids, including lysine, threonine, asparagine and histidine. The increased re-methylation of homocysteine and subsequent availability of methionine may have led to increased protein synthesis, as methionine is typically considered the primary limiting amino acid in diets formulated for cats [21, 95]. In growing animals, including kittens and various livestock species, supplementing dietary choline or betaine has repeatedly increased protein deposition, as observed through improved lean mass gains and/or carcass quality [96–99]. Similarly, insulin-resistant mice deficient in CTP:phosphoethanolamine cytidylyltransferase deficient (Pcyt2) had increased protein synthesis and turnover within skeletal muscle with choline supplementation [100]. As body composition was not assessed in the present study, it is unclear whether choline supplementation improved lean muscle mass in these cats. Lysine is also a major component of carnitine [101]. Therefore, it is unclear whether the increase in serum lysine may be representative of the increased concentration of circulating carnitine. Additionally, if acetyl-CoA production was increased due to augmented fatty acid oxidation, it may have reduced the requirement of these amino acids to be converted to various Kreb's cycle intermediates, such as oxaloacetate, a-ketoglutarate, and succinyl-CoA [102]. However, without assessing enzyme activity within the Kreb's cycle, this cannot be determined.

In addition to being oxidized to betaine, dietary choline can be incorporated into PC through the CDP-choline pathway (also known as the Kennedy pathway). Alternatively, PC can also be produced by the donation of three SAMe methyl groups to phosphatidylethanolamine (PE) through the PE N-methyltransferase (PEMT) pathway, [35]. In the present study, serum PC increased with choline at 6 x NRC RA (Fig 3). Similarly, concentrations of LPC increased at 6 x NRC RA. This is not a surprising finding as LPC is produced by the cleavage of PC by phospholipase A$_2$ (PLA$_2$) [103]. This data parallels previous increases in serum PC and LPC in growing kittens supplemented with additional dietary choline [48]. As previously hypothesized, the increase in PC with choline at 6 x NRC RA was likely responsible for the higher serum lipid and lipoprotein concentrations previously reported in these same overweight cats [47]. The increased serum lipid and lipoprotein profiles suggested increased hepatic lipid mobilization. The supplementation of PC in rats with induced hepatic steatosis reduced hepatic TAG synthesis and accumulation [104]. Similarly, rabbits with hyperlipidemia had decreased concentrations of hepatic cholesterol and TAG following dietary PC supplementation through soy lecithin [105]. Assessing concentrations of PC within hepatocytes may

therefore be suggested when determining the choline status of an animal. Serum SM and HSM concentrations also increased in the present study with choline at 6 x NRC RA (Fig 3). Similar to PC, SM is a choline-containing phospholipid [106]. The essential roles of SM and HSM within the body include supporting cell membrane structure and function, and SM is also a major constituent of the myelin sheath [107]. Although increased SM is often correlated with obesity and metabolic syndrome in humans [108], there is no data to support whether the same is true for cats. Valtolina et al. [109] found increased plasma and hepatic concentrations of SM in cats diagnosed with FHL. However, it remains unclear what the role of SM is in the development and progression of FHL.

In addition to being converted into PC or betaine, dietary choline can also result in the production of TMAO by the gut microbiota [110, 111]. In the present study, serum TMAO concentrations only increased at the highest dose of choline at 8 x NRC RA. Given that choline-derived metabolites, such as PC and betaine, increased further with the 6 x NRC RA dose, as opposed to the 8 x NRC RA dose, it is possible that the ability of the small intestine to absorb dietary choline may have been saturated, resulting in greater production of TMAO [112, 113]. A similar dose-response relationship has previously been established in rats [113]. Extensive reviews have been published discussing the causal relationship between increased circulating TMAO and increased risk of kidney and cardiovascular diseases [114–116]. However, the exact mechanisms are not clear. To the authors' knowledge, the health implications of TMAO in cats have not yet been studied. Given that cats are obligate carnivores whose diets would include a higher inclusion of TMAO precursors, such as choline and carnitine, it is unclear whether the risks of increased circulating TMAO are the same for cats.

A limitation of the current study was that fasted serum metabolites were only collected at one time point (at the end of each treatment period). Therefore, the data presented herein only captures one specific moment of an otherwise dynamic physiological state. Additionally, the serum metabolite pool is comprised of multiple different biochemical pathways from different tissues which function together. In the future, to complement metabolomic profiles, the expression of genes and enzymes that function within and regulate these pathways should be assessed, to better understand which pathways may be up- or down-regulated with additional choline supplementation. Furthermore, although certain metabolites showed statistically significant concentrations between doses, understanding the biological relevance in terms of health benefits or risks is difficult to determine. Additionally, the use of male cats in the present study was the result of previous work investigating choline supplementation following gonadectomy in kittens fed to mimic *ad libitum* feeding [48, 99]. Male cats have previously been found to be at greater risk of obesity as compared to females [5, 13, 117, 118]. Although there appear to be no differences in endogenous choline synthesis in cats based on sex and gonadectomy in cats [119], healthy female cats have been reported to have lower concentrations of plasma and hepatic TAG, but the same concentrations of circulating PC, as compared to male cats [109]. Therefore, differences in hepatic lipid mobilization with additional choline supplementation between female and male cats may exist. Although women have been found to have lower betaine-homocysteine- methyltransferase (BHMT) activity, as compared to men [120], the impact on circulating sex steroids and differences between sexes on one-carbon metabolism in cats has not been studied and may also change the lipotropic effects of choline dose in cats.

Overall, choline at up to 8 times the NRC RA was well-tolerated by cats and presented no adverse health outcomes. The highest dose of choline at 8 times the published RA by the NRC may have overcome the capacity for choline absorption by the small intestine, as supported by increases in serum TMAO. However, the biological relevance of this is unknown in cats. Choline has become a nutrient of interest in the maintenance of hepatic health and FHL

prevention in overweight and obese cats during energy restriction, due its roles in one-carbon metabolism, phospholipid biosynthesis, and β-oxidation through L-carnitine [75, 121]. Overweight male adult cats consuming dietary choline at 6 and 8 times the NRC RA had increased concentrations of one-carbon metabolites, suggesting an improvement in methyl status with increased dietary choline intake. Additionally, concentrations of PC increased with a dietary choline intake of 6 times RA, further supporting previous hypotheses of improved hepatic lipid packing and mobilization via VLDL in overweight and obese cats [46, 47]. Although the cats in the present study were fed at maintenance energy requirements, increasing body condition score can result in greater hepatic TAG and greater risk of insulin resistance [49, 122]. Taken together, these results support the use of dietary choline for the maintenance of hepatic health. Additionally, choline may support lean muscle mass in overweight and obese cats. Protein synthesis may have been improved in overweight cats, as suggested by increased serum concentrations of numerous amino acids with choline intake at the two highest doses. Future studies are warranted to investigate the supplementation of dietary choline at 6 times the NRC RA for cats at risk of FHL, such as obese cats undergoing weight loss. These studies should consider including an evaluation of body composition as well as the expression of proteins and genes in hepatic, adipose and muscle tissue involved in the regulation of metabolic processes. This will help elucidate the impact of choline on protein synthesis, fatty acid oxidation, lipolysis and/or lipogenesis.

## Acknowledgments

Thank you to the Ontario Veterinary College Pet Nutrition team and student volunteers for their help with blood collection for this study, and their assistance with cleaning and providing social enrichment to the cats. We would also like to thank Cara Cargo-Froom for helping with the dietary amino acid analyses for this study, and Michelle Edwards for her assistance with the statistical analysis.

## Author Contributions

**Conceptualization:** Alexandra Rankovic, Anna K. Shoveller, Marica Bakovic, Gordon Kirby, Adronie Verbrugghe.

**Data curation:** Alexandra Rankovic, Anna K. Shoveller, Marica Bakovic, Adronie Verbrugghe.

**Formal analysis:** Alexandra Rankovic, Anna K. Shoveller, Marica Bakovic, Gordon Kirby, Adronie Verbrugghe.

**Funding acquisition:** Marica Bakovic, Gordon Kirby, Adronie Verbrugghe.

**Investigation:** Alexandra Rankovic, Hannah Godfrey, Caitlin E. Grant.

**Methodology:** Alexandra Rankovic, Hannah Godfrey, Caitlin E. Grant, Anna K. Shoveller, Marica Bakovic, Gordon Kirby, Adronie Verbrugghe.

**Project administration:** Alexandra Rankovic, Anna K. Shoveller, Adronie Verbrugghe.

**Resources:** Anna K. Shoveller, Adronie Verbrugghe.

**Supervision:** Anna K. Shoveller, Marica Bakovic, Gordon Kirby, Adronie Verbrugghe.

**Validation:** Alexandra Rankovic, Anna K. Shoveller, Marica Bakovic, Gordon Kirby, Adronie Verbrugghe.

**Visualization:** Alexandra Rankovic, Adronie Verbrugghe.

**Writing – original draft:** Alexandra Rankovic.

**Writing – review & editing:** Alexandra Rankovic, Hannah Godfrey, Caitlin E. Grant, Anna K. Shoveller, Marica Bakovic, Gordon Kirby, Adronie Verbrugghe.

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
