## [Decision Letter · Decision Letter 0]

19 Oct 2022

PONE-D-22-21927Serum metabolomic analysis of the dose-response effect of dietary choline in overweight cats fed at maintenance energy requirementsPLOS ONE

Dear Dr. Verbrugghe,

Thank you for submitting your manuscript to PLOS ONE. After careful consideration, we feel that it has merit but does not fully meet PLOS ONE’s publication criteria as it currently stands. Therefore, we invite you to submit a revised version of the manuscript that addresses the points raised during the review process.

We look forward to receiving your revised manuscript.

Kind regards,

Mikhail Y. Golovko, PhD

Academic Editor

PLOS ONE

Journal Requirements:

Additional Editor Comments:

Your manuscript was reviewed by two experts in the field and the comments are enclosed. Although both reviewers provided overall positive feedback, the concerns from both reviewers should be addressed in the revision. I would specifically emphasize the comment regarding “The one lean cat was enrolled to balance BW between groups but was not included in the statistical analysis.”. In addition, the authors should provide a link to the full data set that is available for review without restrictions.

Reviewers' comments:

Reviewer's Responses to Questions

**Comments to the Author**

1. Is the manuscript technically sound, and do the data support the conclusions?

Reviewer #1: Yes

Reviewer #2: Yes

2. Has the statistical analysis been performed appropriately and rigorously? 

Reviewer #1: Yes

Reviewer #2: Yes

3. Have the authors made all data underlying the findings in their manuscript fully available?

Reviewer #1: Yes

Reviewer #2: Yes

4. Is the manuscript presented in an intelligible fashion and written in standard English?

Reviewer #1: Yes

Reviewer #2: Yes

5. Review Comments to the Author

Reviewer #1: This study is investigating the role of choline in overweight cats.

Overall, I think this is a well written study that requires some minor revisions.

Firstly, I think that the title of the manuscript should reflect that the authors did this work in male cats. Within the manuscript adult cats should be revised to male adult cats.

I think there should also be a statement made in the discussion, as well. Do the authors have any data on female cats or ideas on how the data would look in those animals?

The sentence below is not clear.

Page 12: Choline dose did not impact 261 any of the other one-carbon metabolites, including choline, creatine, formic acid and L-carnitine 262 (P Dose = 0.874, 0.432, 0.353, and 0.116, respectively).

Within the results section can the authors add a brief statement of why each test was conducted?

The authors discuss the impact of choline supplementation increasing one-carbon metabolites. Do they think this would impact other diseases such as cancer? There have been studies investigating the role of folic acid supplementation on cancer.

Reviewer #2: The authors should clarify why “The one lean cat was enrolled to balance BW between groups but was not included in the statistical analysis.” If it’s not included in the analysis should it also not be included in the BW calculations?

In Table 1 and other tables, it would be good to show the number of cats in each group.

The authors are suggesting that choline supplementation may prove useful in the prevention of FHL. However, it is not clear if the cats are at risk for FHL besides being obese. This should be clarified a bit. The authors show different metabolites changing levels in different choline concentrations, however, it is not clear in the manuscript how choline will help prevent FHL when FHL was not shown in these cats. A clarification would be helpful.

6. PLOS authors have the option to publish the peer review history of their article (what does this mean?). If published, this will include your full peer review and any attached files.

Reviewer #1: No

Reviewer #2: No

---

## [Author Response · Author response to Decision Letter 0]

1 Dec 2022

The rebuttal letter including responses to reviewer comments has been uploaded as a separate document.

---

## [Decision Letter · Decision Letter 1]

9 Jan 2023

Serum metabolomic analysis of the dose-response effect of dietary choline in overweight male cats fed at maintenance energy requirements

PONE-D-22-21927R1

Dear Dr. Verbrugghe,

We’re pleased to inform you that your manuscript has been judged scientifically suitable for publication and will be formally accepted for publication once it meets all outstanding technical requirements.

Kind regards,

Mikhail Y. Golovko, PhD

Academic Editor

PLOS ONE

Additional Editor Comments (optional):

Reviewers' comments:

Reviewer's Responses to Questions

**Comments to the Author**

1. If the authors have adequately addressed your comments raised in a previous round of review and you feel that this manuscript is now acceptable for publication, you may indicate that here to bypass the “Comments to the Author” section, enter your conflict of interest statement in the “Confidential to Editor” section, and submit your "Accept" recommendation.

Reviewer #1: All comments have been addressed

Reviewer #2: All comments have been addressed

2. Is the manuscript technically sound, and do the data support the conclusions?

Reviewer #1: Yes

Reviewer #2: Yes

3. Has the statistical analysis been performed appropriately and rigorously? 

Reviewer #1: Yes

Reviewer #2: Yes

4. Have the authors made all data underlying the findings in their manuscript fully available?

Reviewer #1: Yes

Reviewer #2: Yes

5. Is the manuscript presented in an intelligible fashion and written in standard English?

Reviewer #1: Yes

Reviewer #2: Yes

6. Review Comments to the Author

Reviewer #1: (No Response)

Reviewer #2: My comments have been addressed, thank you. The authors have addressed my main concerns and added the clarifications I requested in the discussion section.

7. PLOS authors have the option to publish the peer review history of their article (what does this mean?). If published, this will include your full peer review and any attached files.

Reviewer #1: No

Reviewer #2: No

---

## [Editor Report · Acceptance letter]

13 Jan 2023

PONE-D-22-21927R1 

Serum metabolomic analysis of the dose-response effect of dietary choline in overweight male cats fed at maintenance energy requirements 

Dear Dr. Verbrugghe:

I'm pleased to inform you that your manuscript has been deemed suitable for publication in PLOS ONE. Congratulations! Your manuscript is now with our production department. 

Kind regards, 

on behalf of

Dr. Mikhail Y. Golovko 

Academic Editor

PLOS ONE